EMBO
Molecular Medicine

# An AI-assisted morphoproteomic approach is a supportive tool in esophagitis-related precision medicine

Sven Mattern[1,2], Vanessa Hollfoth[1], Eyyub Bag[1], Arslan Ali[1], Philip Riemenschneider[3], Mohamed A Jarboui [4], Karsten Boldt[4], Mihaly Sulyok[1], Anabel Dickemann[1], Julia Luibrand[1], Stefano Fusco [2,5], Mirita Franz-Wachtel[6], Kerstin Singer[1], Benjamin Goeppert[7,8], Oliver Schilling[9,10], Nisar Malek[2,5], Falko Fend[1], Boris Macek [6], Marius Ueffing [4] & Stephan Singer [1,2]✉

## Abstract

Esophagitis is a frequent, but at the molecular level poorly characterized condition with diverse underlying etiologies and treatments. Correct diagnosis can be challenging due to partially overlapping histological features. By proteomic profiling of routine diagnostic FFPE biopsy specimens (n = 55) representing controls, Reflux- (GERD), Eosinophilic-(EoE), Crohn's-(CD), Herpes simplex (HSV) and Candida (CA)-esophagitis by LC-MS/MS (DIA), we identified distinct signatures and functional networks (e.g. mitochondrial translation (EoE), immunoproteasome, complement and coagulations system (CD), ribosomal biogenesis (GERD)), and pathogen-specific proteins for HSV and CA. Moreover, combining these signatures with histological parameters in a machine learning model achieved high diagnostic accuracy (100% training set, 93.8% test set), and supported diagnostic decisions in borderline/challenging cases. Applied to a young patient representing a use case, the external GERD diagnosis could be revised to CD and ICAM1 was identified as highly abundant therapeutic target. This resulted in CyclosporinA as a personalized treatment recommendation by the local multidisciplinary molecular inflammation board. Our integrated AI-assisted morphoproteomic approach allows deeper insights in disease-specific molecular alterations and represents a promising tool in esophagitis-related precision medicine.

**Keywords** Esophagitis; FFPE-Proteomics; Precision Medicine; Mass Spectrometry; Targeted Therapy
**Subject Categories** Computational Biology; Digestive System
Published online: 3 February

## Introduction

Esophageal inflammation is a common finding in patients suffering from dysphagia or retrosternal pain. Patients with esophagus-related symptoms frequently undergo endoscopic evaluation with mucosal biopsy sampling and subsequent histological evaluation. The etiology of esophageal inflammation can be roughly categorized into reflux-esophagitis (gastro-esophageal reflux disease, GERD) and non-reflux-associated entities (Maguire and Sheahan, 2012). The prevalence varies depending on the etiology and is assumed to be as high as 20% for the most common cause, which is reflux-esophagitis (Dellon and Hirano, 2018; Dent et al, 2005). Among the non-reflux related entities are allergic, autoimmune, and infectious diseases such as eosinophilic esophagitis (EoE), Crohn's disease (CD) manifestation in the esophagus, and herpes simplex virus (HSV) or candida (CA) infection (Alsomali et al, 2017; Mastracci et al, 2020; Muir and Falk, 2021). To assign the adequate treatment for each patient (e.g., proton pump inhibitors (PPI), glucocorticoids, immune-modulatory biologics, or anti-infectious therapy) a precise diagnosis is crucial, which can be challenging by conventional histological evaluation alone due to overlapping histomorphological features (Lam et al, 2022; Mohamed et al, 2019; O'Donnell and Krishnan, 2022).

GERD shows mostly non-specific findings including epithelial changes such as hyperplasia of the basal cell layer, papillary elongation as well as spongiosis. Inflammation, which is not always present, usually consists of neutrophilic granulocytes (especially related to erosions and ulceration), as well as eosinophils (Eos) and monocytes (Tripathi et al, 2018). Accordingly, even for a single institute an interobserver disagreement between experienced pathologists is estimated with 10% (Allende and Yerian, 2009) and in non-erosive reflux disease (NERD) histology alone appears insufficient as a diagnostic tool (Collins et al, 1989).

[1]Department of Pathology and Neuropathology, University of Tübingen, Tübingen, Germany. [2]Center for Personalized Medicine (ZPM), Tübingen, Germany. [3]University Cancer Center Frankfurt (UCT), University of Frankfurt, Frankfurt, Germany. [4]Core Facility for Medical Proteomics, Institute for Ophthalmic Research, Center for Ophthalmology, University of Tübingen, Tübingen, Germany. [5]Department of Internal Medicine I, University of Tübingen, Tübingen, Germany. [6]Proteome Center Tübingen, University of Tübingen, Tübingen, Germany. [7]Institute of Pathology and Neuropathology, Hospital RKH Kliniken Ludwigsburg, Ludwigsburg, Germany. [8]Institute of Tissue Medicine and Pathology, University of Bern, Bern, Switzerland. [9]Institute of Pathology, University Medical Center Freiburg, Faculty of Medicine – University of Freiburg, Freiburg, Germany. [10]Center for Personalized Medicine (ZPM), Freiburg, Germany. ✉E-mail: Stephan.Singer@med.uni-tuebingen.de

EoE also displays spongiosis and infiltrates of eosinophilic granulocytes, but per definition, the latter are more pronounced compared to GERD (current threshold: >15 intraepithelial eosinophils per high power field (HPF)) (Thakkar et al, 2023). The histomorphological changes can be longitudinal, patchy, and even more severe in the proximal biopsies in contrast to the mainly distal manifestation of GERD (Gonsalves and Aceves, 2020; Haggitt, 2000). Thus, depending on the number of biopsies taken, the sensitivity of histological EoE diagnosis could be as low as 55% (Gonsalves et al, 2006). In addition, cases of pauci-eosinophilic esophagitis (<15/HPF) have been described (Hui et al, 2017) and represent a particular diagnostic challenge.

Upper gastrointestinal (GI) involvement in CD is, in general, considered to be underdiagnosed ranging between 5% (adults) and 30–50% (children), with increasing detection rates due to a more extensive diagnostic workup (Abuquteish and Putra, 2019; Kovari and Pai, 2022). Esophageal CD manifestations include infiltrates of neutrophilic and eosinophilic granulocytes, lymphocytes/ plasma cells and histiocytes, erosions and ulcerations as well as epithelioid granulomas in a subset of cases. While the latter are considered relatively specific, their occurrence is ranging between 0 and 25% (Pimentel et al, 2019) indicating that the majority of cases are prone to be misdiagnosed.

Esophagitis due to HSV infection typically affects immunocompromised patients. It often shows ulceration and a mixed inflammatory infiltrate. The infected epithelium can exhibit cytopathic alterations, including ground glass nuclei, multinucleation, and so-called Cowdry type A inclusion bodies. However, these alterations are not always visible in biopsies, because of the uneven distribution of infected cells. Therefore, immunohistochemistry (IHC) or PCR are frequently required for validation (Genereau et al, 1997; Rajasekaran et al, 2021).

The large majority of all infectious esophagitis cases are caused by *Candida albicans*. The diagnosis is made by identification of pseudohyphae in the PAS-stain. The histological changes of the epithelium as well as the inflammatory infiltrate are non-specific (Alsomali et al, 2017).

Given the considerable overlap of morphological features of relevant esophagitis types additional techniques and approaches (in combination with conventional histology) are required to increase diagnostic accuracy and guide appropriate treatment.

Mass spectrometry-based proteomics, particularly liquid chromatography, and tandem mass spectrometry (LC-MS/MS), emerged as a powerful tool in molecular biology over the last decades and allows the identification and quantification over several thousand proteins in a given sample (Aebersold and Mann, 2016). Recent advances in sample preparation protocols rendered also formalin-fixed and paraffin-embedded (FFPE) tissue samples (even in small amounts) amenable to LC-MS/MS (Buczak et al, 2020; Buczak et al, 2018). FFPE represents the standard tissue processing of routine diagnostic samples in all pathology institutes worldwide. With routine diagnostic biopsies being now sufficient for FFPE-LC-MS/MS this technique holds great potential for broad applications in precision diagnostics/medicine. MS proteomic analyses have already been performed in the context of different inflammatory diseases such as rheumatoid arthritis, psoriasis, and inflammatory bowel disease (Arafah et al, 2020; Han et al, 2012; Nanni et al, 2009; Park et al, 2021; Ren et al, 2021; Rochman et al,

2023; Sobolev et al, 2022). However, the vast majority of them did not use FFPE tissue, including a recent publication by Rochman et al (Rochman et al, 2023) focusing on EoE using fresh frozen tissue samples. To the best of our knowledge, this is the first study to apply FFPE-LC-MS/MS in combination with machine learning to routine diagnostic esophageal biopsies to analyze EoE, GERD, CD, and infectious esophagitis types in a comparative manner to identify specific proteomic alterations for each diagnosis, and to deepen our understanding of the underlying pathophysiological mechanisms and therapeutic implications.

## Results

For the present study, a cohort of routine diagnostic FFPE esophageal biopsies ($n = 55$) obtained during upper GI endoscopy was assembled. The detailed composition of the cohort is presented in Fig. 1A and further explained in the Methods section. The patient samples used for the following normal vs. disease comparisons (illustrated in Figs. 2–6) are categorized under the "Discovery approach"/"References" group (Fig. 1A). For representative histological images, see Fig. EV1. Patient characteristics are summarized in Dataset EV1A. Histological characteristics were assigned according to the predefined categories as detailed in Dataset EV1B. Each case was re-evaluated by experienced pathologists regarding the respective diagnosis and representative areas (including squamous epithelium, stroma, and inflammatory cells), which were marked for macrodissection. Protein extraction, digestion, and peptide purification was performed based on an established protocol (Buczak et al, 2020; Buczak et al, 2018). For in depth proteomic characterization of the samples a label free high-resolution LC-MS/MS approach using an Orbitrap Tribrid Fusion mass spectrometer was chosen (operated in data-independent acquisition (DIA) mode). Figure 1B,C illustrate the workflow including the analysis steps resulting in a total of >7000 identified protein groups, which were filtered according to quality criteria (see Methods section) resulting in a total of 5554 protein groups across all non-infectious esophagitis and control samples ($n = 44$, Dataset EV1C) to perform further analyses. Gene and protein names are listed in Dataset EV1D.

### Proteomic profiling of EoE cases suggests distinct changes related to proinflammatory cytokines IL5, IL4, and IL13 as well as to mitoribosomal proteins

First, we analyzed proteomic alterations in EoE in comparison with the control samples (discovery cohort). Based on a log2 fold change (FC, disease/control) $\geq 1/\leq -1$ and an adjusted (permutation-based FDR) $p$-($= q$)-value <0.05 cut-off 589 differentially abundant proteins (DAPs) could be identified (Fig. 2A (highlighted in red and blue, respectively), Dataset EV2A). Among the top ten highest abundant proteins (Fig. 2A, highlighted in dark red and labeled), there were several markers for eosinophilic granulocytes such as EPX, CLC, RNASE2, RNASE3, PRG2, and PRG3 building a functional network by using the STRING database (Fig. 2B). This finding was in full agreement with an allergic immune response involving IL5-mediated recruitment of eosinophils (Kouro and Takatsu, 2009) and corresponds to the histological hallmark of an

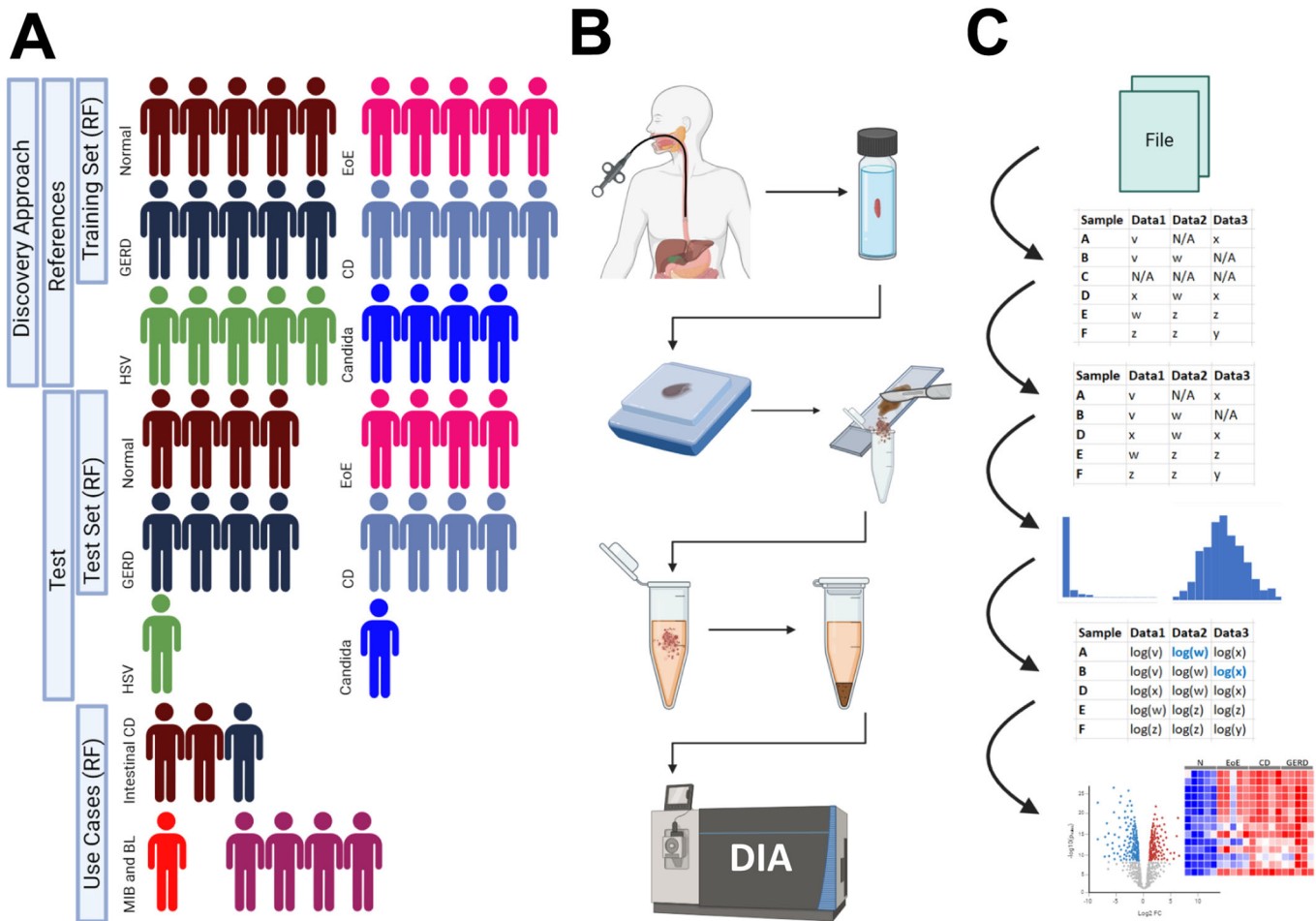

**Figure 1. Patient cohort composition, workflow of the FFPE-LC-MS/MS approach, and analysis process of the acquired data.**

(A) Composition of the patient cohorts and usage in the respective analyses throughout the study. (B) Tissue acquisition via endoscopic biopsy, formalin fixation, paraffin embedding (FFPE), macrodissection, protein extraction, protein precipitation (not shown: digestion and cleanup), liquid chromatography-tandem mass spectrometry (LC-MS/MS; DIA-mode), and data analysis. (C) Major steps of the data analysis process: conversion of raw data, filtering according to the quality criteria, log transformation, data imputation, and statistical testing. All images were created using biorender. The heatmap was created using Morpheus.

eosinophils-dominated infiltrate in these patients. Accordingly, based on the top ten highest abundant protein enrichment analyses using METASCAPE yielded "Eosinophilic Esophagitis" as the most significant human disease term (DisGeNet) with a $p$ value of $10^{-14}$ (Fig. 2C), proving the plausibility of our proteomic approach. Besides IL5, also IL4 and IL13 play an important role in chronic allergic diseases, including EoE (Avlas et al, 2023). We therefore tested if protein/gene set members of the corresponding term "Interleukin-4 and Interleukin-13 Signaling (Reactome Pathway (RP))" are enriched among DAPs in EoE using GSEA (https://www.gsea-msigdb.org/) (Subramanian et al, 2005). Indeed, as shown in the enrichment plot in Fig. 2D (left panel) there was a significant ($p$ value = 0.006) overrepresentation of DAPs related to this pathway (Enrichment Score (ES) = 0.58, Normalized Enrichment Score (NES) = 1.61). Interestingly, members of this pathway/network are connected via ALOX5 to an associated functional network containing ALOX15, LTA4H, and GPX4, representing the top-ranked enrichment term "Biosynthesis of E-series 18(R)-

resolvins" (RP, STRING database) (Fig. 2D right panel; Dataset EV2B).

Importantly, we also could confirm a subset of markers of a previously published transcriptomic diagnostic panel for EoE (EDP) (Wen et al, 2013) (Fig. EV3A; Dataset EV2C). For instance, higher abundance for ALOX15, ANO1, CA2, CDH26, EPB41L3, GCNT3, GLDC, POSTN, and TNFAIP6 as well as lower abundance of EPB41L3 exclusively in EoE could be detected. We also observed increased levels of CLC, COL1A2, CTSC, EPPK1, and EPX and lowered levels of ALOX12, ENDOU, and GRPEL2, however, these changes were not EoE-specific in our patient cohort and also occurred in CD and/ or GERD (Fig. EV3A; Dataset EV2C). Another subset of markers of the EDP panel was not significantly changed at the proteomic level such as PTGFRN, ARG1, and FKBP5 (Fig. EV3A; Dataset EV2C). We assume that besides the influence of different cohort composition/size, not necessarily all significantly altered transcripts correspond to exclusive DAPs in EoE.

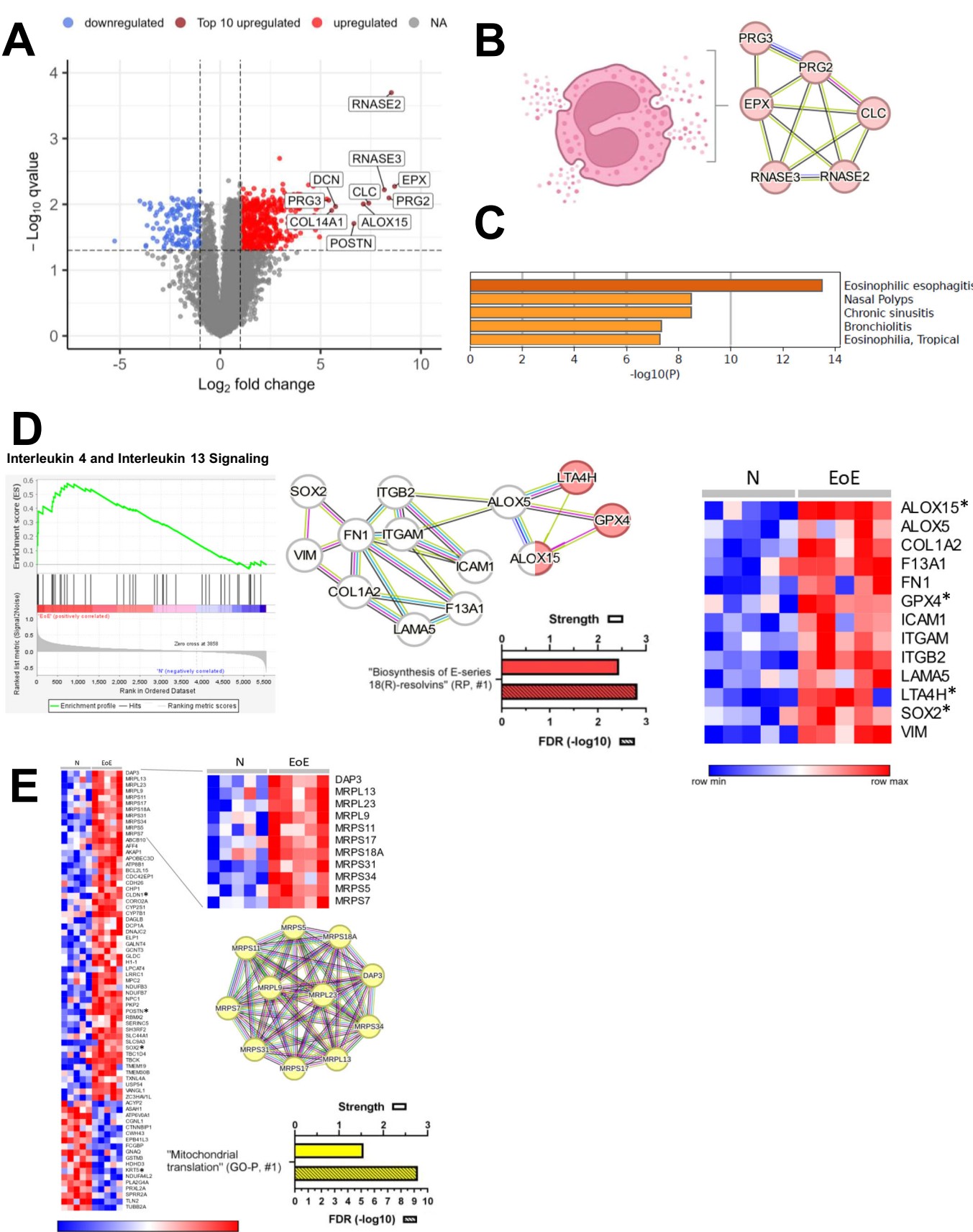

◄ **Figure 2. Proteomic profiling of EoE cases suggests distinct changes related to proinflammatory cytokines IL5, IL4, and IL13.**

(A) Volcano plot showing differentially abundant proteins (DAPs) with higher (red) and lower (blue) abundance in EoE compared to the control samples (N). Top 10 of highest abundant proteins are labeled and highlighted in dark red. (B) Eosinophilic markers form a network displayed by proteins as circles and their respective connections via lines with different color coding (STRING database). (C) DisGeNet analysis via Metascape of the top 10 highest abundant proteins in EoE shows "Eosinophilic esophagitis" as the top hit. (D) GSEA analysis of "Interleukin-4 and Interleukin-13 Signaling" (RP) displaying a significant enrichment in this term (enrichment score (ES): 0.58, Normalized Enrichment Score (NES): 1.61, Nominal p value: 0.006). The STRING network of this term is also connected to another network representing "Biosynthesis of E-serin 18(R)-resolvins" (RP) via ALOX15 (ALOX5 is also part of this term and differentially but not exclusively expressed in this comparison), asterisk (*) marks DAPs exclusive for the comparison NvEoE. (E) Heatmap shows (to the best of our knowledge) previously unrecognized EoE-specific DAPs (except those with an asterisk (*) marks suggested by former reports based on IHC/IF/WB analyses). STRING network of DAPs representing "mitochondrial translation" (GO). Bar diagrams (D, E) show strength and false discovery rate (FDR, −log10, hatched bar) as well as the rank in the respective database. For a complete list of all related enrichment terms see Dataset EV2B, F. The lowest value in each row of the heatmaps is displayed in dark blue, and the highest value in dark red. The scale traverses white. Heatmaps were created using Morpheus. Eosinophilic granulocyte (B) was created using biorender.

Next, we contextualized our results with a recent, elegant study, which included proteomic data generated by LC-MS/MS analyses (DDA) using fresh frozen biopsies from EoE patients (Rochman et al, 2023). While fresh frozen tissue usually allows a deeper proteome coverage than FFPE, we were able to quantify ~5500 proteins (blue ellipse) showing a 97% overlap with the ~2200 proteins (rose hued ellipse) quantified by Rochman et al (Fig. EV3B). This underscores the strength of the preparation protocol combined with the DIA strategy chosen in our study. By applying the FC thresholds of that study (log2FC ≥0.58 (=FC 1.5)/ ≤ −0.58 (=FC −1.5)), we could confirm various consistent sets of DAPs (q < 0.05) related to the biological processes highlighted by the authors, such as "chromosome organization and mitosis" (e.g., MCM2 and MCM4), "epidermal differentiation" (e.g., CRNN, CSTA, and DSC2), "splicing" (e.g., SF3A2, SF3B1, and PRPF8), "eosinophils" (EPX, PRG2, and RNAS3), and "cytoskeleton organization" (ARPC1B, FERMT3, and CORO1A), among others (Fig. EV3C; Dataset EV2D).

Overlapping DAPs between both studies being specific for EoE in our cohort (adapted log2FC ≥ 0.58/ ≤ −0.58, q < 0.05) are marked as exclusively altered in Dataset EV2D (n = 14) including ALOX15. However, besides those overlapping DAPs, we identified an additional 73 exclusive proteins (Fig. 2E and marked in Dataset EV2E). Subjecting these to enrichment analyses (STRING database) revealed mitochondrial ribosomal proteins to be significantly overrepresented with the corresponding top-ranking GO term "Mitochondrial translation" among higher abundant DAPs (Dataset EV2F).

Taken together, we identified proteomic alterations that can be related to highly relevant interleukins in EoE (IL5, IL4, and IL13) and to the biosynthesis of E-series resolvins. Moreover, we could corroborate previous findings and expand the list of previously unrecognized proteomic EoE markers, including a network of mitoribosomal proteins.

## GERD-specific proteomic alterations suggest increased ribosomal biogenesis and altered peroxisomal function as a response to oxidative stress

Next, we compared the proteomic profiles of GERD specimens with the control samples (discovery cohort), which revealed, based on the thresholds log2FC (disease/control) ≥1/ ≤ −1 and q value <0.05 1295 DAPs, as shown in Fig. 3A (highlighted in red and blue respectively) and Dataset EV3A. GERD-related inflammation was characterized by

neutrophilic granulocytes (with the corresponding markers such as CTSG and ELANE) and, to a lesser extent (compared to EoE), by eosinophils (with the corresponding markers EPX, RNASE2, RNASE3, PRG2, and PRG3) (Fig. 3A,B). The latter was also reflected in a lower log2FC of the respective eosinophilic markers in GERD compared to EoE: for the majority of markers ~6 in GERD (Fig. 3A) and ~8.5 in EoE (Fig. 1A). Reactive oxygen species (ROS) production is directly induced by the refluxate-mediated epithelial damage and indirectly by infiltrating immune cells including eosinophils and neutrophils, highlighting the crucial role of ROS in the pathophysiology of GERD (Rieder et al, 2010; Sharma and Yadlapati, 2021). Accordingly, we found protein networks directly related to the enrichment terms "Hydrogen peroxide biosynthetic process" (GO), "Reactive oxygen species biosynthetic process" (GO), and "Superoxide anion generation" (GO) among DAPs in GERD (Fig. EV4A; Dataset EV3B). However, with one exception (DUOX1) these were not exclusively altered in GERD, but also emerged in EoE and/or CD (Fig. EV4A). Focusing only on exclusively higher DAPs in GERD resulted in enrichments terms such as "rRNA processing", "rRNA metabolic process" (GO Process), "snoRNA binding" (GO Function), and "Ribosome biogenesis in eukaryotes" (KEGG) (Fig. 3C; Dataset EV3C).

MMP2, LAMC2, LAMA2, COL5A3, COL7A1, CRTAC1, COL28A1, THBS, HMCN2, and SPARCL1 were identified as significantly enriched in GERD among the exclusively upregulated DAPs, as indicated by the "Extracellular matrix" enrichment term (UniProt Keywords) (Fig. 3D). While this term is relatively broad, the aforementioned proteins highlight GERD-specific features of extracellular matrix (ECM) composition and remodeling (for general ECM alterations, refer to Fig. 5A).

As shown in Fig. 3E, exclusively downregulated in GERD (Dataset EV3D) corresponding to the enrichment term "Peroxisome" (KEGG) included PRDX5 (an antioxidant enzyme), ACOX3 (involved in β-oxidation of branched-chain fatty acids), ABCD3 (a transporter for importing fatty acids into peroxisomes), SCP2 (critical for lipid transfer and metabolism within peroxisomes), and EPHX2 (detoxifying epoxides) presumably indicating peroxismal dysfunction.

We conclude that GERD-related proteomic alterations can be linked to direct and indirect damage by the refluxate including (particularly neutrophilic) granulocyte recruitment and ROS-related processes. Exclusive DAP networks in GERD included increased ribosomal biogenesis, distinct ECM alterations, and decreased peroxisomal components, potentially representing GERD-specific responses to oxidative stress (see also Discussion section).

## Proteomic profiling of esophageal CD can be linked to Interferon-ɣ signaling and indicate a switch from the constitutive proteasome to the immunoproteasome

We then analyzed proteomic alterations in CD esophagitis (discovery cohort) and observed 1414 DAPs compared to the control samples (Fig. 4A; Dataset EV4A). In keeping with the CD histomorphology showing a mixed inflammatory infiltrate we identified higher abundant proteins related to neutrophilic granulocytes (e.g., CTSG and ELANE), eosinophilic granulocytes (e.g., EPX, RNASE2, RNASE3, CLC, and PRG2), lymphocytes and plasma cells (e.g., LSP1, MZB1, IGKC, IGHG4, IGKV3D-20, IGKV1-17, IGKV3-20, and IGHV3-15), mast cells (e.g., CPA3) as well as antigen-presenting cells (e.g., HLA-DRA, HLA-DRB3) (Fig. 4A,B).

Enrichment and network analyses (STRING, 05/2024) of exclusively upregulated DAPs in CD revealed a well-interconnected group of proteins representing the following enrichment terms (among others): "Antigen processing and presentation" (KEGG), "NOD-like receptor signaling pathway" (KEGG), "Interferon alpha/beta signaling" (RP), "Interferon gamma signaling" (RP), and "Complement and coagulation cascade" (KEGG) (Fig. 4C; Dataset EV4B).

Among exclusively low abundant proteins we identified the proteasomal subunits PSMB5 and PSMB6. These together with upregulated PSME1, PSME2, PSMB8, PSMB 9, and PSMB 10 (Fig. 4D, left panel; Dataset EV4A) indicated a switch from the constitutive proteasome to the immunoproteasome, which in turn could be linked to interferon-ɣ and TNF-α signaling (Tubio-Santamaria et al, 2021) (Fig. 4D, right panel).

Next, we tested if calprotectin (CP, composed of S100A8 and S100A9), as a frequently used biomarker for active inflammation in intestinal CD, also shows a higher abundance in CD esophagitis. Surprisingly, we observed a mild, but significant reduction of S100A8 and S100A9 (log2FC −0.79 and −0.78) in esophageal CD with an even more pronounced decrease of both proteins in EoE (log2FC −2.1 and −2.3) and GERD (log2FC −1.3 and −1.4) (Fig. EV4B). An explanation for this unexpected finding is provided in the Discussion section. Interestingly, the related proteins S100A16, S100A14, and CRNN together assigned to the term "S-100/ICaBP type calcium binding domain" (Protein domains SMART) were also downregulated in CD, EoE, and GERD. While these CP-related findings differ from intestinal CD, we could detect a higher abundance of predictive protein markers for anti-TNF-α therapy in CD esophagitis as described in serum of inflammatory bowel disease (IBD) patients such as CRP, ITGAV, APOE, and CLU (Gazouli et al, 2013; Kalla et al, 2021; Kumar et al, 2024) (Fig. EV4C).

Our data suggest that key elements of intestinal Crohn's pathophysiology can either be recapitulated or expanded/modified in esophageal CD manifestations at the proteomic level. These included protein networks related to Type I and II interferons and NOD-like receptor signaling, increased complement system components, immunoproteasome, and previously suggested predictive biomarkers.

## Overlapping proteomic alterations are primarily related to an accumulation of ECM components and impaired epithelial maturation in EoE, CD, and GERD esophagitis

Exclusively altered proteins for each esophagitis type as presented above are of particular relevance, however, we also consider recurrent proteomic alterations across the various diseases to be important. A total of 309 proteins (Dataset EV5A) showed increased abundance in all esophagitis subtypes such as COL1A1, COL3A1, COL4A1, COL5A1, COL6A1, FN1, LAMB2, LAMA5, HSPG2, RAC2, and CAV1 with associated and interconnected enrichment terms "ECM-receptor interaction" (KEGG), "Collagen chain trimerization" (RP), and "Focal adhesion" (KEGG) (Fig. 5A; Dataset EV5B). As shown in Fig. 5B, consistently lower DAPs (n = 77, Dataset EV5A) included KRT4, KRT77, DSC2, KLK12, KLK13, TGM1, TGM3, and EVPL representing enrichments in "Keratinization" (Gene Ontology-Process (GO-P)), "Keratinocyte differentiation" (GO-P) and "Formation of the cornified envelope" (RP), respectively. In addition, we observed lower abundant proteins such as IL1RN, IL36A, IL18 ("Interleukin-1 homologs" (SMART)), MUC21, TMPRSS11B, and PADI1 with documented functions in squamous epithelium homeostasis and integrity (G TEx Consortium, 2013; Rochman et al, 2017; Sachslehner et al, 2021) (Fig. 5B; Dataset EV5C).

These data demonstrate that chronic inflammation of the esophagus (independent of the etiology) leads to the accumulation of collagen I, III, IV, V, and VI, and of proteins related to adhesion structures and cell motility. Moreover, our analyses indicate proteomic alterations related to impaired squamous epithelial maturation and defects of the mucosal barrier being generalizable to all esophagitis types.

## Specific viral and fungal proteins are detectable in HSV- and candida-esophagitis by proteomics

Frequent infectious causes of esophagitis, particularly in immunosuppressed patients, include *Candida albicans* (CA) and herpes simplex virus (HSV). For the comparison of infectious esophagitis samples (HSV: n = 5; CA: n = 4) as diagnosed by H&E and IHC or PAS (Fig. EV2A,B) with the normal biopsies (N: n = 5) we used a viral as well as a fungal (instead of a human) peptide reference database for identifying the respective proteins. By doing so, we were able to identify up to 41 protein(groups) in the HSV samples with different abundances, but not in any of the normal samples (Fig. 6A). The resulting proteins could be assigned to different structural components of the virus such as the capsid (e.g., TRX2, CVC2, and SCP), tegument (e.g., DBP, PAP, and NEC2) and envelope (e.g., gB, gG, and gH) related to different time points during infection (Dai and Zhou, 2018; Rajcani and Durmanova, 2000) (Fig. 6A; Dataset EV6A). The proteomic analysis of the CA esophagitis biopsies revealed 30 proteins being present in all CA samples, but not detectable in any of the control samples (Fig. 6B; Dataset EV6B). These proteins represented predominantly ribosomal subunits, and to a lesser extent, metabolic enzymes (Fig. 6B). A subset of these has a documented role in biological processes such as "Induced by azole treatment or linked to azole resistance" (e.g., ADH1, PDC11, and RPL35) or "Proteins with a role in virulence according to Candida Genome Database" (e.g., YHB1 and ASC1) (Martinez-Lopez et al, 2022). We conclude that FFPE-LC-MS/MS allows detailed identification of important infectious causes of esophagitis, such as HSV and CA including potential markers of therapy resistance.

## An integrated morpho-proteomic machine learning approach achieves high diagnostic accuracy in esophagitis cases

Based on a prinicipal component analyses (PCA) using the acquired proteomic data (Reference Set) a clear separation

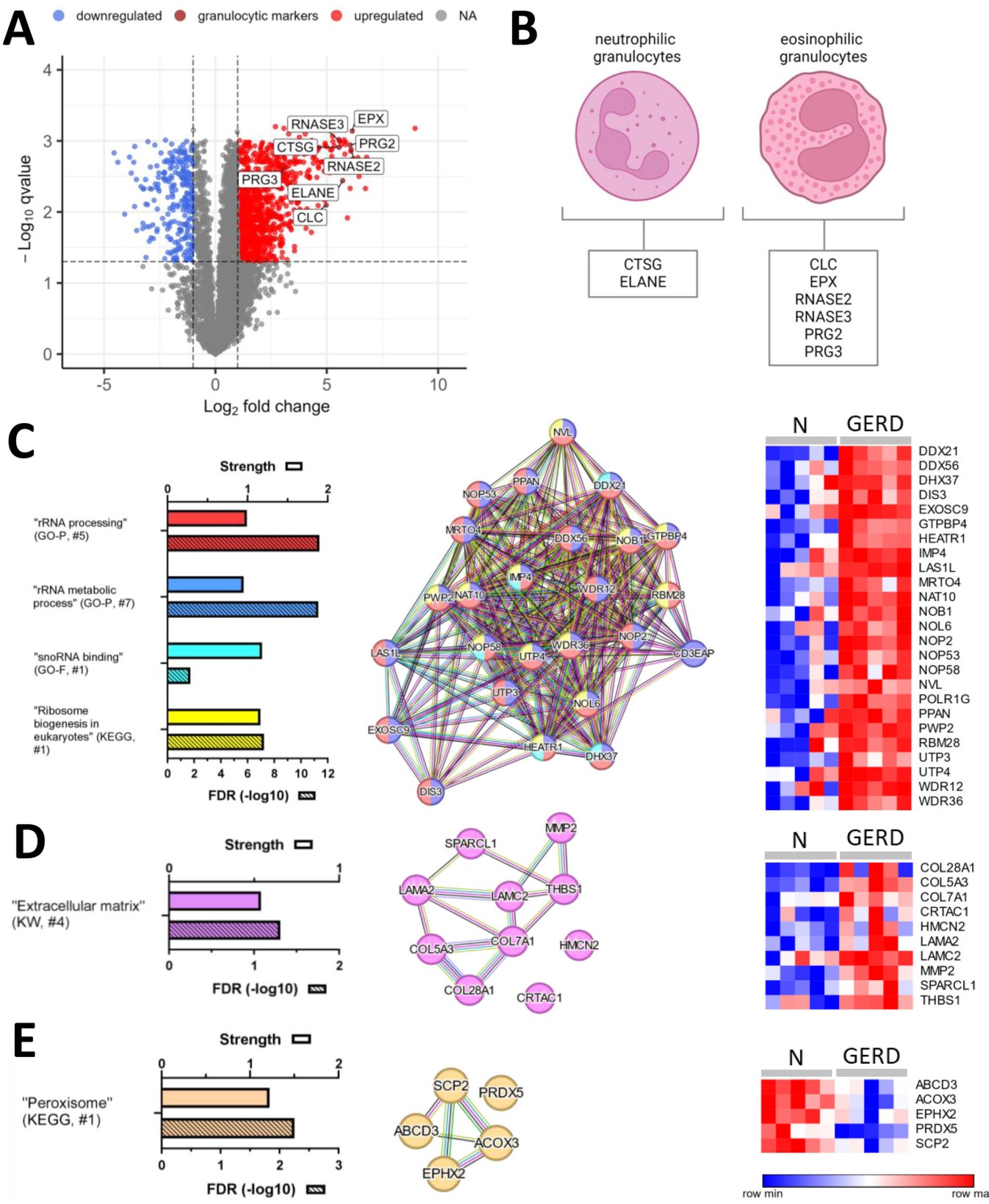

**Figure 3.  GERD-specific proteomic alterations suggest increased ribosomal biogenesis, distinct ECM changes, and altered peroxisomal function as response to oxidative stress.**

(A) Volcano plot showing differentially abundant proteins (DAPs) with higher (red) and lower (blue) abundance in GERD compared to the control samples (N). Markers related to neutrophilic and eosinophilic granulocytes are highlighted in dark red and assigned respectively in (B). (C–E) STRING networks of DAPs and corresponding heatmaps (relative abundances of indicated proteins in each GERD and N sample) as well as a selection of related significant enrichment terms (for a complete list see Dataset EV3C, D) highlighted by bar diagrams with the respective strength and false discovery rate (FDR, −log10, hatched bar) as well as the rank in the respective database. The lowest value in each row of the heatmaps is displayed in dark blue, and the highest value is in dark red. The scale traverses white. Heatmaps were created using Morpheus. Granulocytes (B) were created using biorender. POLR1G = CD3EAP.

particularly between normal and EoE samples was achieved. However, we were not able to discriminate between CD and GERD cases (Fig. EV5). Thus, we employed a supervised machine learning algorithm (Random Forest, RF) integrating proteomic data and histological features for improved disease differentiation (Fig. 7A). We excluded for this analysis the HSV and CA samples, since these can be unequivocally separated by the identification of respective viral and fungal proteins (as demonstrated above and exemplified by an additional HSV (V_HSV1) and candida (V_CA1) test case shown in Fig. EV6A,B). Either histological categorical variables (listed in Dataset EV1A) or the whole proteome or a combination of both was used to train an RF on the aforementioned N, EoE, GERD, and CD samples (n = 20, see also Fig. 1A "Training Set (RF)"). Applied to this training dataset the RF using the proteome alone or combined with the histological features reached 100% accuracy and performed better than solely using the histological variables (accuracy: 95%, mean_sensitivity: 95%, mean_specifity: 98.3%) (Fig. EV6C). Next, we tested the trained RF on a test set (n = 16, Fig. 7B, see also Fig. 1A "Test Set (RF)"). While misclassifying one GERD sample the combined histo-proteomic or pure proteomic RF approach again performed equally better (accuracy: 93.8%, mean_sensitivity: 93.8%, mean_specifity: 97.9%, respectively) compared to the histological features alone (accuracy: 68.8%, mean_sensitivity: 68.8%, mean_specifity: 89.6%) (Fig. EV6D). Interestingly, while the performance of the pure proteomic RF was comparable to that of the combined histo-proteomic RF in our datasets, a subset of histological features emerged as top-ranked variables with the highest importance values in the combined RF approach (Dataset EV7). These features may thereby contribute to better performance in larger patient cohorts.

We then assessed the utility of the combined histo-proteomic RF in aiding diagnostic decision-making for challenging and border-line cases, which are summarized as "use cases" in Fig. 1A. First, inconspicuous esophageal samples of two CD patients (N1(CD) and N2(CD)) and a case of a patient with intestinal CD, but with the histological diagnoses of GERD (GERD(CD)) were analyzed. As shown in Fig. 7C each case was correctly predicted as N or GERD, respectively. Moreover, we also tested borderline (BL) cases of EoE patients (BL1-3) and one borderline GERD case (BL4) with the following characteristics and results. BL1 was histologically characterized by only very few foci of >15 Eos/HPF as well as large areas without significant eosinophilic infiltrates and BL2 showed an increased number of Eos/HPF, however, not exceeding the cut off >15/HPF in any area. Both were predicted as EoE, which we consider the correct diagnosis in the given context (Fig. EV6E, left and middle panel). BL3 presented a rather normal histological appearance with only very few eosinophils (2/HPF) in the biopsy being molecularly analyzed (in contrast to another biopsy

submitted at the same time with clear-cut EoE histology not subjected to proteomic analyses). For BL3, the RF favored EoE, however, with an almost equal predictive value for N (Fig. EV6E, right panel) together with the aforementioned BL cases, suggesting this approach as a useful tool in pauci-cellular EoE. BL4 represented a challenging GERD case from a patient with a complex medical history (intestinal and liver transplantation with associated immunosuppressive medication). Here, the RF predicted GERD only as the second most likely diagnosis (Fig. EV6F) possibly pointing out the limitations of the approach in complex clinical settings including long-standing systemic immunosuppression (Hampton et al, 2008; Stewart et al, 2008).

Finally, we applied the morpho-proteomic RF in the context of a young female patient (with known intestinal CD) presenting with dysphagia and esophageal stenosis, who was discussed in the interdisciplinary molecular inflammation board at the University Hospital Tübingen (UKT) (Fig. 7D, left panel). The patient was externally diagnosed with GERD. However, the UKT gastroenter-ologists suspected rather a CD manifestation in the esophagus, obtained esophageal biopsies and requested a morpho-molecular analysis to (a) clarify the diagnosis and (b) to identify potential drug targets. As demonstrated in Fig. 7D (middle panel), the integrated approach clearly classified the case as CD esophagitis with a predictive value >0.5 so that in conjunction with the clinical setting the external GERD diagnosis was revised accordingly. Moreover, when compared with the other CD cases of our training dataset, we observed a particularly high abundance of ICAM1 (Fig. 7D, right panel), and therefore, CiclosporinA targeting ICAM1 was suggested as a personalized molecular-based treatment option for this patient.

Taken together, our data suggest that the morpho-molecular RF approach does not only confirm clear-cut diagnoses, but also provides support in (re-)classifying borderline/challenging cases.

## Discussion

We could demonstrate in this study the potential of comparative proteomic profiling of different esophagitis subtypes using routine FFPE biopsies in combination with machine learning to address challenges in esophagitis diagnostics and to improve the molecular understanding of these diseases.

In EoE patient samples, we identified proteomic alterations connected to key interleukins in allergic inflammation, such as IL5, IL4, and IL13, and a related functional network assigned to the enrichment term "Biosynthesis of E-series 18(R)-resolvins". This network involved ALOX5, ALOX15, LTA4H, and GPX4 (with LTA4H and GPX4 also being connectable to IL4/IL13 signaling

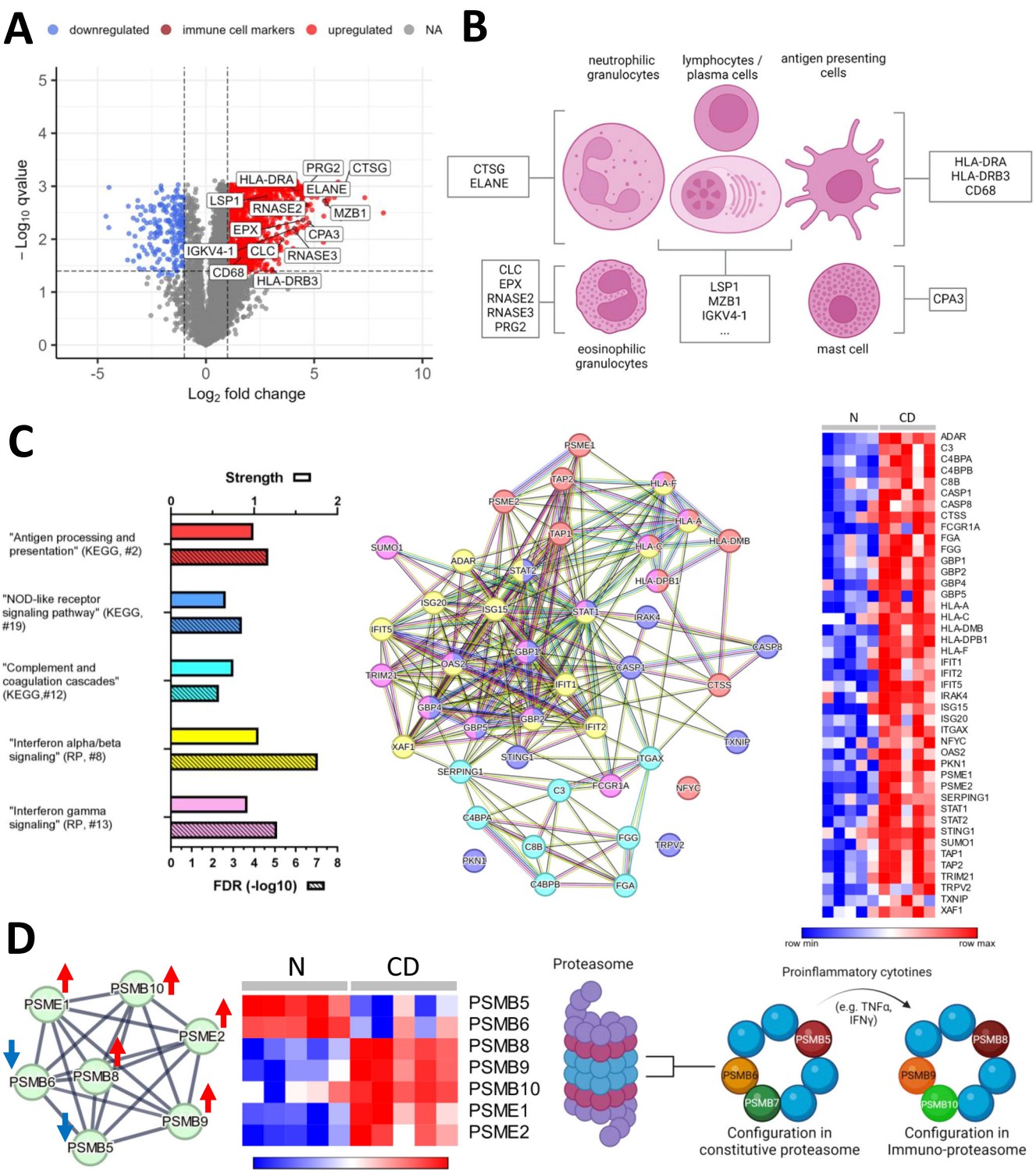

(Agbor et al, 2014; Wenzel et al, 2017; Zaitsu et al, 2000)). E-series resolvins, such as those biosynthesized via ALOX15 are lipid mediators derived from omega-3 fatty acids and exhibit anti-inflammatory properties, potentially contributing to the regulation of allergic inflammation (Basil and Levy, 2016; Isobe et al, 2012; Sastre et al, 2018). Conversely, GPX4 is suggested to have an antagonistic function to ALOX15, probably indicating the presence of a regulatory feedback loop (Ivanov et al, 2015). These findings underline the complex interplay between the IL4/IL13 pathway and E-series resolvin biosynthesis and highlight the potentially divergent roles of enzymes/proteins involved in EoE-related inflammation.

**Figure 4.** Proteomic profiling of esophageal CD can be linked to Interferon-γ signaling, NOD-like receptor signaling, and complement system, and indicate a switch from the constitutive proteasome to the immunoproteasome.

(A) Volcano plot showing differentially abundant proteins (DAPs) with higher (red) and lower (blue) abundance in CD compared to the control samples (N). Markers related to neutrophilic granulocytes, lymphocytes/plasma cells, antigen-presenting cells, mast cells, and eosinophilic granulocytes are highlighted in dark red and assigned respectively in (B). (C, D) (left panel) STRING networks of DAPs and corresponding heatmaps (relative abundances of indicated proteins in each CD and N sample) as well as a selection of related significant enrichment terms (for complete list see Dataset EV4B) highlighted by bar diagrams with the respective strength and false discovery rate (FDR, −log10, hatched bar) as well as the rank in the respective database. (D) (right panel) switch from the constitutive proteasome to the immunoproteasome (modified from Tubio-Santamaria et al, 2021, with additional information from Atkinson et al, 2012 and Rouette et al, 2016). The lowest value in each row in the heatmaps is displayed in dark blue, the highest value in dark red. The scale traverses white. B and D were created using biorender. The heatmaps were created using Morpheus.

Enrichment analyses of the 73 exclusive DAPs in EoE, which, to the best of our knowledge, have not been reported as EoE-specific protein markers before (with the exception of POSTN, SOX2, KRT5, and CLDN1 (Clevenger et al, 2023; Politi et al, 2017; Rochman et al, 2017; Wu et al, 2018)) revealed a network of mitoribosomal proteins (Fig. 2E). Mitochondria, particularly from eosinophils, play a pivotal role in allergic inflammation by a generation of reactive oxygen species (ROS) and regulation of apoptosis, among other processes (Koranteng et al, 2024). Thus, we speculate that higher abundant mitoribosomal proteins possibly contribute to increased ROS production and aberrant prolonged survival of eosinophils, thereby potentially sustaining and amplifying inflammation and subsequent tissue damage in EoE. Depending on the results of further studies detailing mitochondrial translation in EoE, the process of mitochondrial translation and its components could potentially serve as therapeutic targets in the future.

An additional therapeutic perspective in EoE was opened up by Rochman and colleagues, who could mechanistically link the MCM complex to basal zone hyperplasia and suggest MCM-directed therapy as a potential treatment option for EoE (Rochman et al, 2023). Basal zone hyperplasia is, indeed, prominent in EoE, but can also be observed in other types of esophagitis, consistent with the fact that MCMs were also significantly upregulated in our cohort in other types of esophagitis. Consequently, targeting MCMs could potentially be effective in treating other forms of esophagitis as well.

While these findings suggest/confirm promising therapeutic targets and potential diagnostic biomarkers in EoE, it could be considered a limitation of our study to exclusively use mass spectrometry-based proteomics without further validation of key markers and signatures by other techniques. However, we believe that the strong correlation of proteomic alterations in EoE found by us and by Rochman et al (Rochman et al, 2023)—despite differences in starting material (FFPE vs. fresh frozen), EoE patient cohort composition, sample preparation protocols, mass spectrometry instrumentation, and acquisition mode - provides robust validation of our results. Moreover, a subset of highlighted EoE markers, such as ALOX15, has already been shown to be positive in the squamous epithelium by IHC in EoE patients, even with eosinophils <15/HPF (Hui et al, 2017; Matoso et al, 2014. In addition, POSTN, SOX2, KRT5, and CLDN1 have also previously been reported in EoE through alternative detection methods such as IHC, immunofluorescence, or immunoblotting (Clevenger et al, 2023; Politi et al, 2017; Rochman et al, 2017; Wu et al, 2018). Lastly, we found a strong correlation between proteomic alterations and histological features including immune infiltrate composition, epithelial changes, and stromal remodeling.

In GERD patients, neutrophils and eosinophils are recruited by IL8 and PAF released by the mucosa being exposed to gastric fluid (de Vries et al, 2009; Rieder et al, 2010; Sharma and Yadlapati, 2021). These immune cells and the refluxate-mediated epithelial damage are key players in the production of ROS, which in turn are considered crucial for the pathophysiology of GERD (Rieder et al, 2010; Sharma and Yadlapati, 2021). In principle consistent with this concept, our analyses identified ROS-associated proteomic alterations linked to terms such as "hydrogen peroxide biosynthetic process" "reactive oxygen biosynthetic process" and "superoxide anion generation". However, since these alterations were also identified in other types of esophagitis, we hypothesize that GERD-specific responses to oxidative stress might rather be represented by the exclusive DAPs. In this context, we could link upregulated exclusive DAPs to all key steps of rRNA processing and ribosomal biogenesis: rRNA Transcription (POLR1G), rRNA Modification (e.g., NOP2 and NAT10), processing of rRNA (e.g., DDX21, DDX56, NOB1, and EXOSC9), assembly and maturation of the 60S (e.g., GTPBP4 and NOP53) and 40S (UTP3, UTP4, and PWP2) subunit (Uniprot 06/24). Notably, these changes did not include a concomitant higher abundance of ribosomal proteins, which may indicate defective ribosomal assembly and impaired ribogenesis. Interstingly, ROS are more commonly reported to inhibit rRNA synthesis and ribosomal biogenesis, particularly in acute settings (Szaflarski et al, 2022). However, there are instances where oxidative stress results in increased/hyperactive ribosome biogenesis (Hattori et al, 2014; Huang et al, 2021), depending on its duration, severity, and biological context. GERD is typically characterized by chronic and recurring exposure to the refluxate resulting in varying levels of oxidative stress. This stress pattern being further modulated by the intake of PPIs may contribute to the phenotype of increased and potentially dysfunctional ribosome biogenesis.

Among downregulated exclusive DAPs in GERD, we identified peroxisomal proteins such as PRDX5, ACOX3, ABCD3, SCP2, and EPHX2, which are involved in the regulation of oxidative stress and lipid metabolism. These changes can be interpreted as a sign of peroxisomal dysfunction, which in turn may enhance ROS accumulation, impair peroxisomal lipid metabolism, and subsequently increase cellular damage and inflammation. While dysfunctional peroxisomes triggering an immune response due to disrupted fatty acid metabolism and elevated ROS have been primarily studied in neuroinflammation and -degeneration (Di Cara et al, 2019; Di Cara et al, 2023), similar effects may be relevant in GERD as well.

A subset of proteins associated with the extracellular matrix (e.g., MMP2, LAMC2, and COL7A1) were shown to be significantly enriched in GERD. This observation may also be

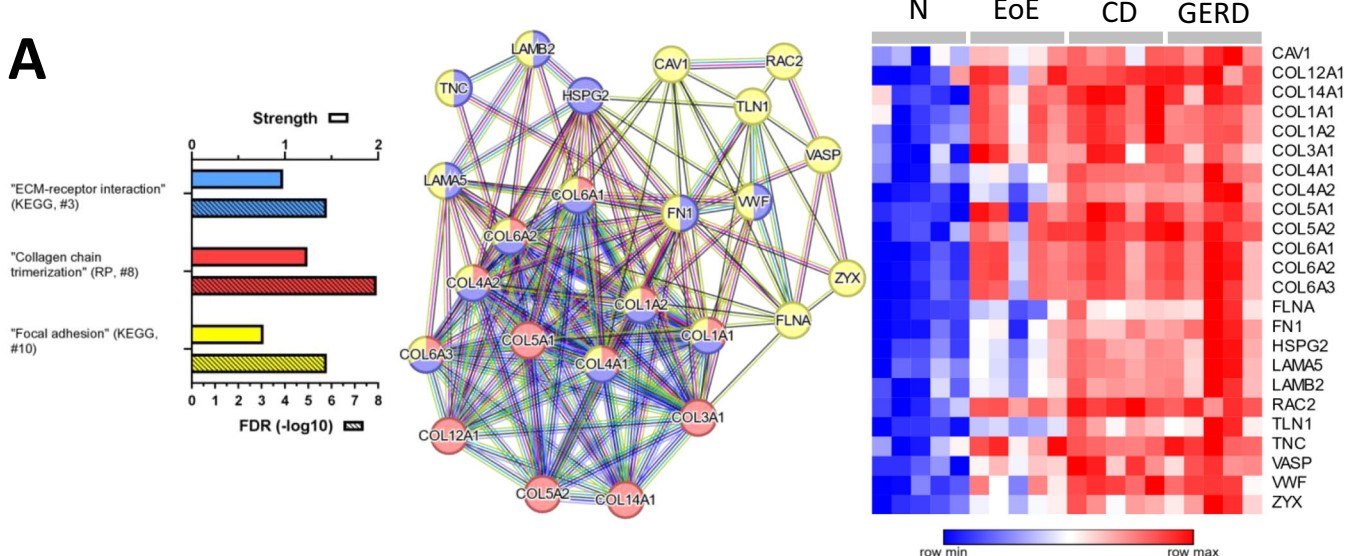

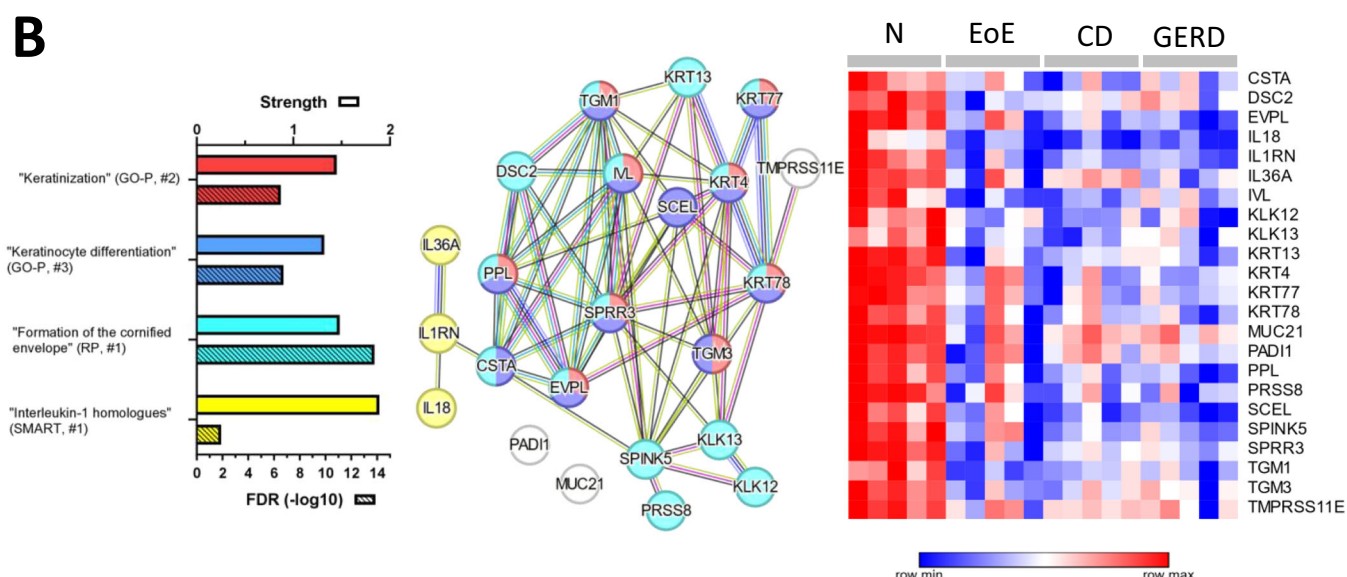

**Figure 5. Proteomic alterations common to all esophagitis types compared to control samples.**

(A, B) STRING networks of DAPs and corresponding heatmaps (relative abundances of indicated proteins in each N, EoE, CD, and GERD sample) as well as a selection of related significant enrichment terms (for a complete list see Dataset EV5B,C) highlighted by bar diagrams with the respective strength and false discovery rate (FDR, -log10, hatched bar) as well as the rank in the respective database. (B) White nodes are single proteins that are not part to the illustrated term (based on the STRING database), but also show functionality in keratinization (based on literature). The lowest value in each row in the heatmaps is displayed in dark blue, the highest value in dark red. The scale traverses white. The heatmaps were created using Morpheus.

associated with GERD-relevant cytokines IL1β, IL8, PAF, and ROS, which are partially interconnected. For instance, IL1β induces in fibroblasts not only collagen production and MMP2 expression, but also IL8 secretion, which in turn recruits neutrophils releasing ROS finally resulting in perpetuated inflammation and progressive fibrosis (Gabasa et al, 2020; Kolb et al, 2001; Larsen et al, 1989; Postlethwaite et al, 1988). Additionally, ROS can induce fibrosis via TGFβ activation (Latella, 2018)

including production of LAMC2 (Olsen et al, 2003), and COL7A1 (Mauviel et al, 1994). PAF is known to bind to the PAF receptor (PAFR) on fibroblasts, further promoting fibrosis through collagen accumulation and other ECM components (Correa-Costa et al, 2014; Latchoumycandane et al, 2015). Collectively, while fibrosis/ECM accumulation is a generalizable phenomenon across all esophagitis subtypes it may still exhibit subtype-specific components.

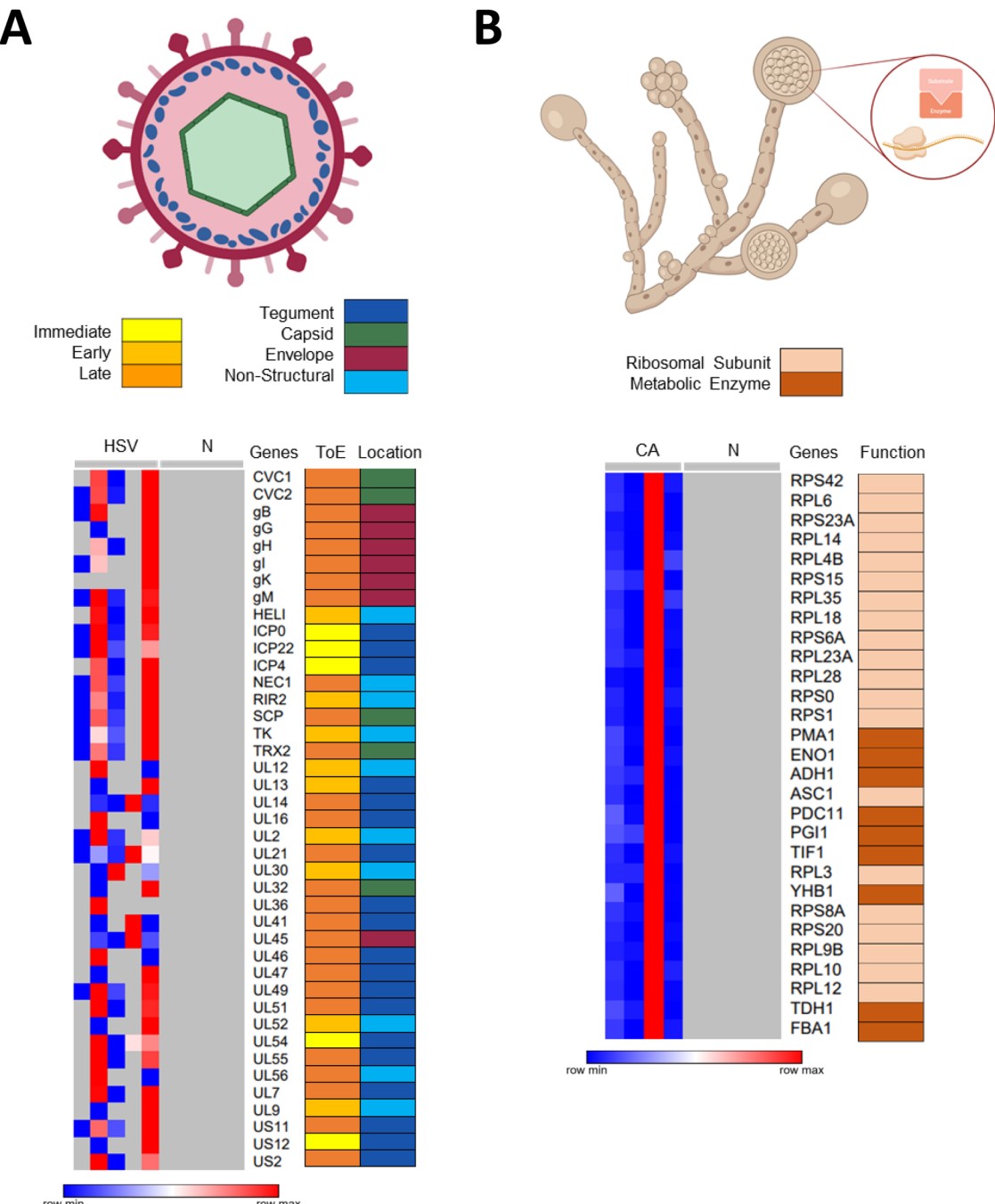

**Figure 6. Identification of HSV-specific and CA-specific proteins in respective infectious esophagitis samples.**

(A) Heatmap (left) displays indicated HSV-specific proteins detected in HSV esophagitis samples, but not in the control samples. The time of expression (ToE) as well as the corresponding viral structure (tegument, capsid, and envelope) for each protein is shown (right). (B) Heatmap (right) illustrates *Candida albicans* (CA)-specific proteins detected in samples with histopathological evidence of CA esophagitis, but not in the control samples. The function of the identified proteins is displayed on the right. The lowest value in each row of the heatmaps is displayed in dark blue, and the highest value in dark red. The scale traverses white. Missing values are gray. Heatmaps were created using Morpheus. The illustrations of the virus particle (A) and the fungal structures (B) were created using biorender.

Exclusive protein networks identified in esophageal CD could be linked to processes and pathways, such as interferon signaling, with a documented role in inflammatory bowel disease (Andreou et al, 2020; Cummings et al, 2010). For instance, a subset of related transcripts such as STAT1, STAT2, ISG15, ISG20, and OAS2 was reported as upregulated in intestinal biopsies of IBD patients in full accordance with our proteomic findings (Ostvik et al, 2020). Moreover, a CD is preferentially characterized by a Th1 cytokine profile including Interferon-γ and TNF-α. Interferon-γ stimulation results in the replacement of the constitutive catalytic proteasome

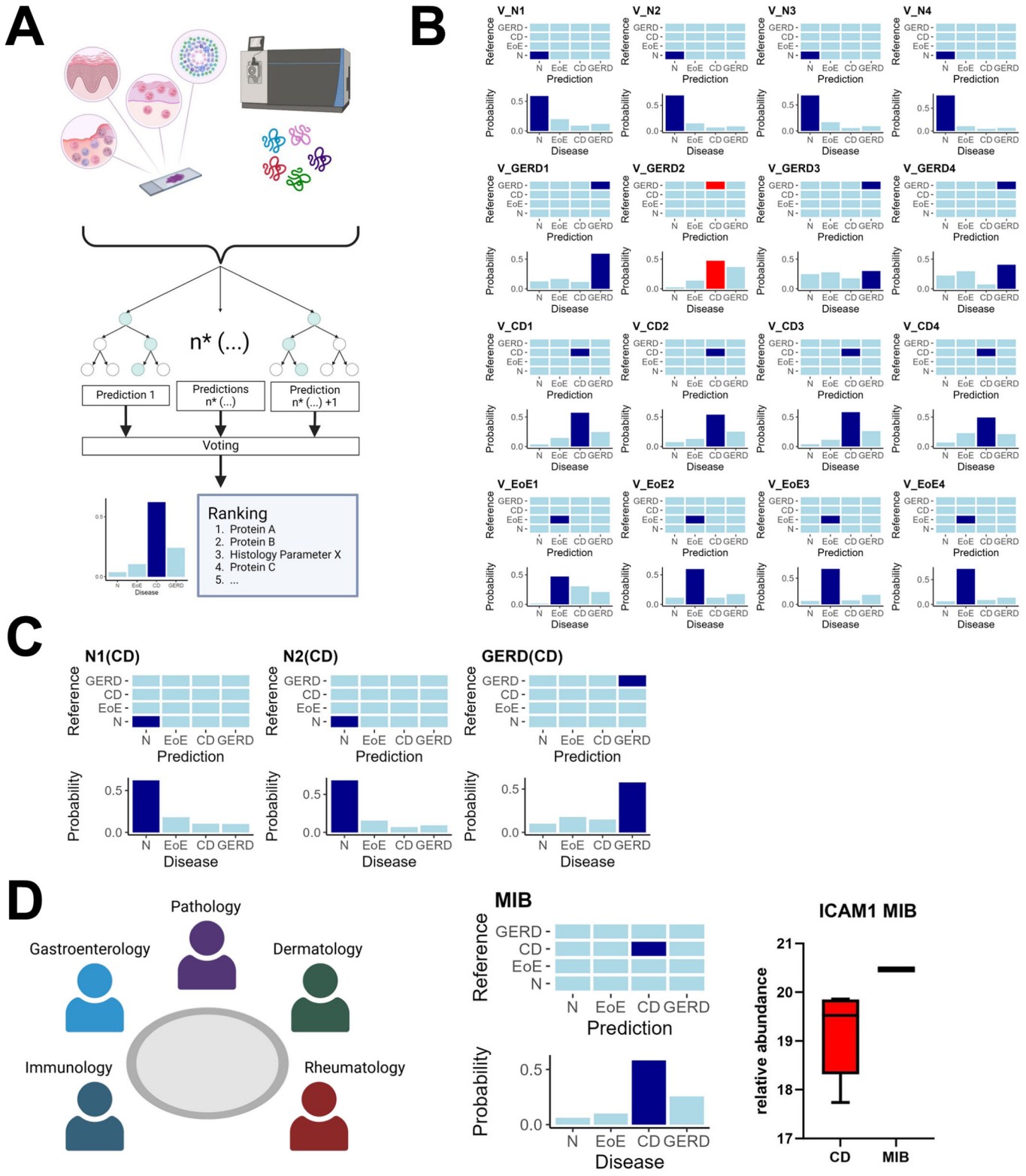

**Figure 7. A combined morpho-proteomic and machine learning approach for esophagitis diagnostics.**

(A) Histomorphological parameters, as listed in Dataset EV1A, as well as the whole proteome, were integrated in a random forest (RF) machine learning model. (B) Bar diagrams illustrate the RF predictive values for each test case of N, GERD, CD, and EoE. Squares in the confusion matrices and bars colored in blue indicate the correct classification for each case by the RF, with red markings indicating an incorrect classification. (C) Testing of samples with known intestinal Crohn's disease but no histological or clinical evidence of esophageal Crohn's disease manifestation to avoid overcalling. (D) Use case of the morpho-molecular approach in a patient presented in the molecular inflammatory board (MIB, left panel), who was externally diagnosed as GERD, but proved to be CD (middle panel). As a druggable target, the patient showed a high abundance of ICAM1 (20.5) compared to other CD samples ($n = 5$). The median (19.53) is shown as the center, minimum (17.74), and maximum (19.86) are indicated by whiskers and 25% quartile (18.33) and 75% quartile (19.85) by bounds of the box.

subunits PSMB6 (b1), PSMB7 (b2), and PSMB5 (b5) by the inducible subunits PSMB 9(b1i), PSMB 10 (b2i), and PSMB8 (b5i), respectively, forming the so-called immunoproteasome (Atkinson et al, 2012; Rouette et al, 2016; Tubio-Santamaria et al, 2021). Incorporation of these subunits is necessary for the production of a variety of MHC class-I restricted T cell epitopes (Abi Habib et al, 2022; Ferrington and Gregerson, 2012). This remodeling of the proteasome was clearly evident based on our proteomic analyses in esophageal CD and fits well with observations made in intestinal CD. On the other hand, we were surprised to find discrepant observations between esophageal and intestinal CD in terms of calprotectin (CP) levels. As opposed to a high abundance of CP in intestinal CD, we detected a decrease of CP in CD esophagitis (and also in EoE and GERD). We assume that inflammation-associated epithelial damage in CD, GERD, and EoE interferes with the normal CP production of intact squamous epithelium (Jukic et al, 2021; Pawar et al, 2015), which is characterized by relative high amounts of CP in contrast to intestinal epithelium. The major source of intestinal/fecal CP, however, are neutrophilic granulocytes (Zittan et al, 2018). Thus, we assume that CD exhibiting the most pronounced neutrophilic infiltrates compared to EoE and GERD shows the strongest partial compensation of reduced CP production by the squamous epithelium. As a result, the decrease in CP levels (though still a decrease) relative to normal controls is mildest in esophageal CD when compared to EoE and GERD.

By applying a morpho-molecular RF approach, we could not only confirm clear-cut EoE, CD, and GERD (conventional) diagnoses, but also provides support in (re-)classifying border-line/challenging cases. Particularly, the latter appears to us as the actual field of its immediate diagnostic application considering the effort compared to a sole conventional histological evaluation. Interestingly, the RF only using the proteomic data showed equal performance like the combined histo-proteomic RF approach, which underlines the capabilities of proteomic profiling in a diagnostic perspective. However, we would not conclude that histology is negligible, because this comparable performance was observed by using histologically thoroughly evaluated samples. In general, no tissue-based molecular diagnostic analyses should be performed without prior histological (re-)evaluation of a given specimen regarding its composition and representativity. Moreover, in daily practice molecular analysis usually follows (not precedes) conventional histopathological diagnostics, which means that the histological information for each case is already available and integrable. Beyond its diagnostic application, the presented MIB case also highlights the potential of FFPE-proteomics to identify potential drug targets for tailored treatment recommendations in personalized medicine. However, future studies with larger patient cohorts and an expanding disease spectrum are required to further

establish AI-assisted morpho-proteomic analyses of inflammatory diseases of the esophagus (and beyond).

## Methods

### Reagents and tools table

| Reagent/resource | Reference or source | Identifier or catalog number |
|---|---|---|
| **Experimental Models** | | |
| FFPE tissue samples from routine diagnostic esophageal biopsies ($n = 55$) | Archive of the Institute of Pathology of the University Hospital Tübingen | Datasets EV1A, C |
| **Antibodies** | | |
| HSV I + II Antibody; poly Rabbit | Medac, Wedel, Germany | #BSB-5641 |
| **Chemicals, enzymes, and other reagents** | | |
| ultra-pure water | New England Biolabs, Ipswich, MA, USA | #B1500L |
| Dithiothreitol | Carl Roth GmbH, Karlsruhe, Germany | #6908.3 |
| Iodoacetamide | Sigma/Merck, Darmstadt, Germany | #I1149-5G |
| Urea | Carl Roth GmbH, Karlsruhe, Germany | #2317.3 |
| HEPES buffer | Carl Roth GmbH, Karlsruhe, Germany | #HN77.3 |
| LysC | FUJIFILM Wako pure chemicals corporation, Osaka, Japan | #125-05061 |
| Trypsin | Promega, Fitchburg, WI, USA | #ADV5111 |
| Trifluoroacetic acid | Biosolve, Dieuze, France | #0020234131BS |
| iRT standards | Biognosys, Schlieren, Switzerland | #1816351 |
| OptiView DAB IHC detection kit | Roche, Rotkreuz, Switzerland | #06396500001 |
| **Software** | | |
| DIA-NN 1.8.1 | Demichev et al, 2020 | |
| Perseus 1.6.15.0 | Max-Planck Institute of Biochemistry | |
| EnhancedVolcano (v1.20.0) R package | Blighe et al, 2024 | |
| VennDiagram (v1.7.3) R package | Chen, 2022 | |

| Reagent/resource | Reference or source | Identifier or catalog number |
|---|---|---|
| R 4.3.0 | The R Core Team, 2023/ open access | |
| **Other** | | |
| TOMO adhesive glass slides | Matsunami Glass Ind., Ltd., Osaka, Japan | TOM-1190 |
| 0.5 ml reaction tubes | Eppendorf, Hamburg, Germany | #0030121023 |
| 1.5 ml reaction tubes | Eppendorf, Hamburg, Germany | #0030120086 |
| 0.2 ml tubes | LLG-Labware, Meckenheim, Germany | #9407515 |
| Waters OASIS HLB μElution plate 30 μm | Waters, Milford, MA, USA | 186001828BA |
| Bioruptor Plus | Diagenode, Liège, Belgium | |
| Concentrator Plus | Eppendorf, Hamburg, Germany | |
| iMark™ Microplate Absorbance Reader | Bio-Rad, Hercules, CA, USA | |
| Thermomixer comfort | Eppendorf, Hamburg, Germany | |
| Orbitrap Tribrid Fusion mass spectrometer | Thermo Fisher Scientific, Waltham, MA, USA | |

## Methods and protocols

### Patient cohort and samples

All samples (total $n = 55$; Fig. 1A) were retrieved from the archive of the Institute of Pathology of the University Hospital Tübingen, Germany. The subcohort for the exploratory approach contained different types of esophagitis: GERD ($n = 5$), CD ($n = 5$), EoE ($n = 5$) as well as control samples showing no overt inflammation or other alterations (Normal, N, $n = 5$). This subcohort ($n = 20$) also served as a training set for the machine learning approach ("Training set (RF)"). For testing the performance of the RF a subcohort ("Test set (RF)") was compiled consisting of 16 samples (GERD ($n = 4$), CD ($n = 4$), EoE ($n = 4$), N ($n = 4$)). Blinding was not applied since supervised machine learning requires labeled data. Moreover, the RF was applied to challenging or borderline cases ("use cases" ($n = 8$): borderline EoE (BL1-3, $n = 3$), borderline GERD (BL4, $n = 1$), CD patients with normal esophageal biopsies (N(CD), $n = 2$), a CD patient with GERD (N(GERD), $n = 1$) and a CD patient discussed in the molecular inflammation board with the external diagnosis GERD (MIB, $n = 1$)). The infectious esophagitis samples comprised five HSV and four CA samples, which, together with the "Training Set RF" samples made up the complete "Discovery Approach/References" subcohort. One additional HSV and CA sample was used for diagnostic testing, which, together with the "Test Set RF" samples constituted the "Test" subcohort. Every sample was evaluated by experienced pathologists (S.S., K.S., S.M., and F.F.). Patient characteristics are summarized in Dataset EV1A.

### Ethics

The study was approved by the local ethics committee of the medical faculty of the Eberhard-Karls-University Tübingen in the context of an amendment to the existing vote 032/2021BO2.

Written informed consent from patients was acquired, and the study conformed to the principles set out in the WMA Declaration of Helsinki and the Department of Health and Human Services Belmont Report.

### Processing of FFPE diagnostic biopsy samples for proteomic analyses

The samples were further processed, including macrodissection, protein extraction, protein precipitation, protein digestion, and peptide cleanup according to a previously published and modified protocol (Buczak et al, 2020), as detailed below.

Macrodissection. FFPE specimen were cut in 5-μm-thick sections and mounted on TOMO adhesive glass slides. For each case an additional 2.5-μm section was prepared and H&E stained. On the H&E slide the area for macrodissection was marked by an experienced pathologist (see above). Then the unstained slides were deparaffinized by incubation in 2x Xylene for 2 min and then 2x EtOH 100% for 2 min before drying.

Afterwards, macrodissection of the marked areas was performed with a scalpel using the H&E as reference. To avoid powder forming small amounts of ultra-pure water were added to keep the blade moist. The tissue was collected in 0.5 ml tubes.

Protein extraction and heat-induced antigen retrieval. About 100 μl dithiothreitol (DTT)-containing extraction buffer (4% SDS, 100 mM DTT in 1 M Tris pH 8.0) was added to each macrodissected sample. The samples were then sonicated for 15 cycles (20 °C, 1 min ON; 30 s. OFF, setting: HIGH) using a Bioruptor Plus followed by heating to 99 °C for 1 h in a PCR machine. These two steps were repeated once, followed by another sonication step.

Iodoacetamide (IAA) was added to a final concentration of 15 mM and the samples were incubated in the dark for 30 min to allow alkylation of free cysteines. The reaction was quenched by the addition of DTT to a final concentration of 10 mM before transferring samples to a fresh 1.5 ml reaction tube.

Protein precipitation. For protein precipitation, 8x the volume (952 μl) of 100% ice-cold acetone was added to each sample, which were then stored at −20 °C overnight. Centrifugation of the samples was performed at 21,000×g and 4 °C for 30 min. Acetone was removed, and the pellets were washed with 500 μl ice-cold 80% acetone and subsequently centrifuged at 4 °C for 10 min. The washing step was repeated, followed by 2 min of centrifugation under the same conditions. The supernatant was discarded and the samples were completely dried for 15 min using a centrifugal vacuum concentrator.

Protein digestion. Dried samples were resuspended in 46 μl of 3 M urea in 100 mM 4-(2-hydroxyethyl)-1-piperazineethanesulfonic acid (HEPES) by using the Bioruptor Plus for five cycles (60 s ON, 30 s OFF, setting HIGH) until the pellet completely dissolved. The protein yield was estimated via Bradford assay according to the manufacturer's specifications and measured absorbance using an iMark™ Microplate Absorbance Reader.

For each sample up to 5 μg of protein were transferred into a fresh 1.5 ml reaction tube and filled up to 40 μl with 3 M urea in 100 mM HEPES. About 0.5 μl of LysC (0.5 μg/μl) were added, and the samples were incubated for 4 h at 37 °C and 600 rpm using the Thermomixer comfort. Afterward, 40 μl ultra-pure water were added to each

sample, followed by 0.5 μl of Trypsin (0.5 μg/μl) and incubation at 4 h at 37 °C and 600 rpm for 16 h. In samples that yielded less than 5 μg of protein, amounts of LysC and Trypsin were adjusted accordingly.

Peptide clean-up.  After acidification with 7.4 μl of 10% trifluoroacetic acid, the samples were desalted using a vacuum assisted solid phase extraction according to the manufacturer's instructions. The samples were collected in 0.2 ml Tubes, dried in the Concentrator Plus (45 °C, AQ, 45 min) and stored at −20 °C until mass spectrometric measurement.

Liquid chromatography-mass spectrometry (LC-MS/MS).  In depth proteomic characterization of the samples a label free high-resolution LC-MS/MS approach using an Orbitrap Tribrid Fusion mass spectrometer was chosen (operated in data-independent acquisition (DIA) mode). Tryptic peptides were loaded onto a μPAC Trapping Column with a pillar diameter of 5 μm, inter-pillar distance of 2.5 μm, pillar length/bed depth of 18 μm, external porosity of 9%, bed channel width of 2 mm and length of 10 mm; pillars are superficially porous with a porous shell thickness of 300 nm and pore sizes in the order of 100 to 200 Å at a flow rate of 10 μl per min in 0.1% trifluoroacetic acid in HPLC-grade water. Peptides were eluted and separated on the PharmaFluidics μPAC nano-LC column: 50 cm μPAC C18 with a pillar diameter of 5 μm, inter-pillar distance of 2.5 μm, pillar length/bed depth of 18 μm, external porosity of 59%, bed channel width of 315 μm and bed length of 50 cm; pillars are superficially porous with a porous shell thickness of 300 nm and pore sizes in the order of 100 to 200 Å by a linear gradient from 2 to 30% of buffer B (80% acetonitrile and 0.08% formic acid in HPLC-grade water) in buffer A (2% acetonitrile and 0.1% formic acid in HPLC-grade water) at a flow rate of 300 nl per min for 85 min. The remaining peptides were eluted by a short gradient of 10 min from 30% to 95% buffer B; followed by 25 min at 2% of buffer B, the total gradient run was 120 min.

Spectra were acquired in DIA-mode using 50 variable-width windows over the mass range 350–1500 m/z. The Orbitrap was used for MS1 and MS2 detection, with an AGC target for MS1 set to $20 \times 10^4$ and a maximum injection time of 100 ms. MS2 scan range was set between 200 and 2000 m/z, with a minimum of 6 points across the peak. Orbitrap resolution for MS2 was set to 30 K, isolation window set to 1.6, AGC target to $50 \times 10^4$, and maximum injection time to 54 ms. MS1 and MS2 data were acquired in centroid mode.

In order to check for retention time (RT) stability, iRT standards were spiked in each sample according to the manufacturer recommendations, and the 11 iRT peptide sequences were manually added to the Human Uniprot FASTA database used during DIA-NN search to generate the precursor ion library used for MS data analysis. The retention times of the iRT peptides were compared by plotting against their linear RT to assess the quality of the LC-MS runs and the performance of the DIA-NN search. To reduce the possibility of carry over and cross contamination between the samples, two BSA washes (5 fmol/injection) were used between samples, and a trap column wash followed by two BSA washes was used every 10 samples sequence. LC column equilibration using BSA washes reduces non-specific binding and peptide adsorption in the flow path within the LC system, thereby minimizing carry-over.

The above-mentioned workflow is schematically displayed in Fig. 1B.

### Data analysis and statistics

MS raw data files of all samples except for HSV and CA samples (n = 44, Dataset EV1C) were analyzed using DIA-NN 1.8.1 (Demichev et al, 2020) in library-free mode against the human database (UniProt release March 2024, 20412 proteins). First, a precursor ion library was generated using FASTA digest for library-free search in combination with deep learning-based spectra prediction. An experimental library generated from the DIA-NN search was used for cross-run normalization and mass accuracy correction. Only high-accuracy spectra with a minimum precursor FDR of 0.01, and only tryptic peptides (two missed tryptic cleavages) were used for protein quantification. The match between runs option was activated and no shared spectra were used for protein identification. Similarly, Normal, HSV, and CA samples were searched against reviewed entries of HSV1 (taxonomy id 10298, 125 entries), HSV2 (taxonomy id 10310, 95 entries), and C. albicans (taxonomy id 5476, 1412 entries) downloaded on 04.03.2024, in addition to the human database.

The generated quantifications were further analyzed with Perseus (Tyanova et al, 2016). The reference samples were grouped according to their diagnosed disease and protein groups were filtered for at least 60% detected values in at least one group. Imputation was performed on the whole dataset using the normal distribution of 1.8 standard deviations down shift and with a width of 0.3 of the total data matrix. Statistical test was done via two-tailed student's t-test and permutation-based FDR (<0.05) to adjust for multiple test. The data preparation is illustrated in Fig. 1C.

Data visualization and enrichment analyses.  Volcano plots for the different comparisons were generated with the EnhancedVolcano (v1.20.0) package in R (Blighe et al, 2024), Venn diagrams were created using the VennDiagram package in R (v1.7.3) (Chen, 2022). Heatmaps were created using the Morpheus tool (Broad Institute, https://software.broadinstitute.org/morpheus/).

To perform enrichment analysis different online tools with their underlying databases were used. For network analysis and visualization, the STRING Database was used in its most recent iteration (https://string-db.org/, v 12.0.) (Szklarczyk et al, 2019). Proteins are displayed as colored nodes. The edges are color-coded depending on the type of protein association (known interactions: cyan: from curated databases, pink: experimentally determined, predicted interactions: Green: gene neighborhood, Red: gene fusions, Dark Blue: gene co-occurrence, Others: Light green: textmining, Black: co-expression, Light blue: protein homology).

Additionally, the Metascape platform (Zhou et al, 2019) was used —together with the associated databases —for further analysis of underlying pathways, interactions and corresponding disease terms (DisGeNet (https://www.disgenet.org/, v7.0, Integrative Biomedical Informatics Group GRIB/IMIM/UPF) (Pinero et al, 2020), Gene Ontology (GO) (Ashburner et al, 2000; Gene Ontology et al, 2023), Kyoto Encyclopedia of Genes and Genomes (KEGG) (Kanehisa, 2019; Kanehisa et al, 2023; Kanehisa and Goto, 2000), and Reactome Pathways (RP) (Griss et al, 2020)), as well as Protein Domains SMART (Letunic et al, 2021) and UniProt(UniProt, 2023)).

All STRING outputs considered statistically significant according to the default settings: FDR <0.05, strength ≥0.01, and minimum

## The paper explained

### Problem

Esophagitis is a very frequent condition with different etiologies (e.g., chemical damage, autoimmune, allergic, and infectious inflammation). Correct diagnosis of the respective esophagitis type is crucial due to vastly different therapeutic consequences. However, partially overlapping features in endoscopy and histology can result in a diagnostic challenge. Moreover, despite its frequency esophagitis is poorly characterized at the molecular, particularly proteomic level.

### Results

By applying tandem mass spectrometry (LC-MS/MS) to routine diagnostic esophageal biopsies, our study provides disease-specific proteomic signatures of gastro-esophageal reflux (GERD), eosinophilic esophagitis (EoE), esophageal Crohn's disease manifestation (CD), and herpes simplex esophagitis (HSV). The obtained proteomic profiles and differentially abundant proteins (DAPs) not only confirm but also expand current pathophysiological concepts by revealing previously unrecognized aspects of the aforementioned esophagitis types. For EoE IL-5/-4/-13-related DAPs such as CLC, RNASE2, ALOX15, LTA4H, and, unexpectedly, numerous mitoribosomal components were identified. GERD-exclusive DAPs were linked to increased ribosomal biogenesis, distinct ECM alterations, and peroxisomal dysfunction, suggesting a GERD-specific response to oxidative stress directly and indirectly induced by the refluxate. Exclusive DAPs related to antigen processing and presentation, NOD-like receptor signaling pathway, complement and coagulation system, and interferon-γ signaling emerged in CD, with the latter being reflected by a proteasome to immunoproteasome switch (e.g., PSMB5/6/8/9/10). For HSV and CA, pathogen-specific proteins were detectable, including viral proteins derived from the tegument, capsid, and envelope (e.g., ICP22, gH, and SCP) and fungal proteins possibly involved in therapy resistance (e.g., RPL35, ADH1, and PDC11). Proteomic alterations shared by all chronic esophagitis subtypes were related to fibrosis (e.g., accumulation of collagens Type I, III, IV, V, and VI), impaired squamous epithelial maturation, and defects of the mucosal barrier as generalizable responses to esophageal inflammation. Moreover, by combining the respective proteomes and histomorphological parameters in a machine learning approach (Random Forest (RF)) 100% accuracy in the training ($n = 20$) and 93.8% in the test set ($n = 16$) could be reached as well as diagnostic support in borderline cases. Applied to one case of the local molecular inflammatory board the external diagnosis of therapy-refractory GERD could be revised to CD and a potential drug target (ICAM1) was identified as high abundant resulting in CiclosporinA as a personalized treatment recommendation.

### Impact

Our morpho-proteomic RF approach may open up new avenues for improved diagnostics and identification of individual therapeutic targets in esophagitis-related precision medicine. The study also provides a rich source of inflammation-related proteomic data reflecting a broad range of etiologies. Finally, by using routine diagnostic biopsies the general concept of the AI-assisted morpho-molecular approach is transferable to virtually all diseases of the gastrointestinal tract (and beyond) involving histology-based diagnostics.

count in a network: 2) are listed in the datasets (Datasets EV2B,F, 3B–D, 4B, 5B,C). Only a small subset of those significantly enriched terms was selected for illustration in the respective main and EV figures and for further discussion. Criteria for this selection included: the network/term was at least within the top 20 scoring hits (based on the strength value with respective ranks being indicated by numbers (e.g., "#1")), plausible in the given disease-specific context,

and could ideally be further supported by a focused literature search. Thus, the collection of highlighted terms/networks is by no means intended to be exhaustive.

The GSEA v4.3.3 (Broad Institute, https://www.gsea-msigdb.org/) was used to determine the overrepresentation of gene set members/proteins of "Interleukin-4 and Interleukin-13 Signaling (RP)" that was chosen from the Molecular Signature Database (MSigDB; http://www.broadinstitute.org/gsea/msigdb/index.jsp). The default values for the GSEA software parameters were utilized with the proteomics data ($=$ DAPs ($q < 0.05$ and $log2FC \leq -1/ \geq 1$) in the normal vs EoE comparison) as input. Using 1000 random permutations of the phenotypic labels, the statistical significance of the normalized enrichment score (NES) corresponding to the aforementioned gene set was evaluated. A cut-off value of p-value < 0.05 was employed to evaluate the statistical significance of the estimates (Djomehri et al, 2020).

Machine learning. A random forest classification model was fitted using the ranger (Wright and Ziegler, 2017) and caret (Kuhn, 2008) packages in R. The input data consists of ~5300 protein quantities and seven histological features. The forest was trained on the training dataset and tested on the test dataset. Optimal parameters for node size, number of variables randomly selected as candidates for splitting a node and splitting rule were selected with three times repeated fourfold cross-testing trying to achieve the best accuracy. Performance metrics and confusion matrices for training and test datasets were calculated and illustrated with the ggplot2 (v3.5.0) R package (Wickham, 2016).

All calculations were performed with R 4.3.0 (The R Core Team, 2023). With a custom script available in the Expanded View (Code EV1).

### Histology and immunohistochemistry

All samples were re-evaluated by experienced pathologists and selected histological characteristics were assigned according to the predefined categories (Dataset EV1B). These descriptive features per se do not represent a specific diagnosis. For CA cases PAS-reactions were performed to better highlight fungal structures. For all HSV esophagitis cases, an HSVI + II immunohistochemical staining was performed using the HSV I + II Antibody (Dilution 1:175). Antibodies were detected based on the OptiView DAB IHC protocol. The procedure included: 4 min deparaffinization at 72 °C, washing with EZ Prep, incubation with Cell Conditioner No.1 for 40 min at 100 °C, incubation with OV PEROX IHBTR for 4 min, treatment with primary antibody at 37 °C for 32 min, incubation with OV HQ UNIV LINKR for 8 min, incubation with OV HRP MULTIMER for 8 min. The samples were then incubated with OV DAB and OV H2O2 for 8 min, followed by a 4 min incubation with OV Copper. The Counterstaining was performed for 20 min with hematoxylin and for 8 min with BLUING REAGENT afterward. Before mounting the cover slip the slides were taken through an ascending alcoholic series.

# Data availability

The mass spectrometry proteomics datasets generated during the current study are available in the ProteomeXchangeConsortium with the reference number PXD052961 (https://proteomecentral.proteomexchange.org/cgi/GetDataset?

ID=PXD052961) and MassIVE repository with the reference number MSV000094966 (https://massive.ucsd.edu/ProteoSAFe/dataset.jsp?accession=MSV000094966).

The source data of this paper are collected in the following database record: biostudies:S-SCDT-10_1038-S44321-025-00194-7.

## Peer review information

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

## Acknowledgements

We thank Yasmin Bauer and Bianca Köhler as well as Silke Wahl and Anke Biedermann from the Proteome Center Tübingen (PCT) and Franziska Klose (Core Facility for Medical Proteomics) for excellent assistance and support. We thank Ciara-Luna Barnitzke for her support in clinic-pathological data collection and preparation. This work was supported by funds from the state of Baden-Wuerttemberg within the Centers for Personalized Medicine Baden-Wuerttemberg (ZPM).

## Author contributions

**Sven Mattern**: Data curation; Formal analysis; Investigation; Visualization; Methodology; Writing—original draft; Project administration; Writing—review and editing. **Vanessa Hollfoth**: Data curation; Formal analysis; Investigation; Visualization; Methodology; Writing—original draft; Writing—review and editing. **Eyyub Bag**: Data curation; Software; Formal analysis; Validation; Investigation; Visualization; Methodology. **Arslan Ali**: Data curation; Formal analysis; Writing—review and editing. **Philip Riemenschneider**: Data curation; Software; Validation; Investigation. **Mohamed Ali Jarboui**: Data curation; Software; Formal analysis; Validation; Investigation; Writing—review and editing. **Karsten Boldt**: Software; Formal analysis; Supervision; Validation. **Mihály Sulyok**: Software; Formal analysis; Validation; Writing—review and

editing. **Anabel Dickemann**: Data curation; Investigation. **Julia Luibrand**: Investigation. **Stefano Fusco**: Resources. **Mirita Franz-Wachtel**: Resources; Data curation. **Kerstin Singer**: Investigation. **Benjamin Goeppert**: Resources; Validation. **Oliver Schilling**: Resources; Validation. **Nisar, Peter Malek**: Supervision. **Falko Fend**: Investigation. **Boris Macek**: Resources; Data curation; Supervision. **Marius Ueffing**: Resources; Data curation; Supervision. **Stephan Singer**: Conceptualization; Supervision; Funding acquisition; Investigation; Methodology; Writing—original draft; Project administration; Writing—review and editing.

Source data underlying figure panels in this paper may have individual authorship assigned. Where available, figure panel/source data authorship is listed in the following database record: biostudies:S-SCDT-10_1038-S44321-025-00194-7.

## Funding

## Disclosure and competing interests statement
The authors declare no competing interests.

# Expanded View Figures

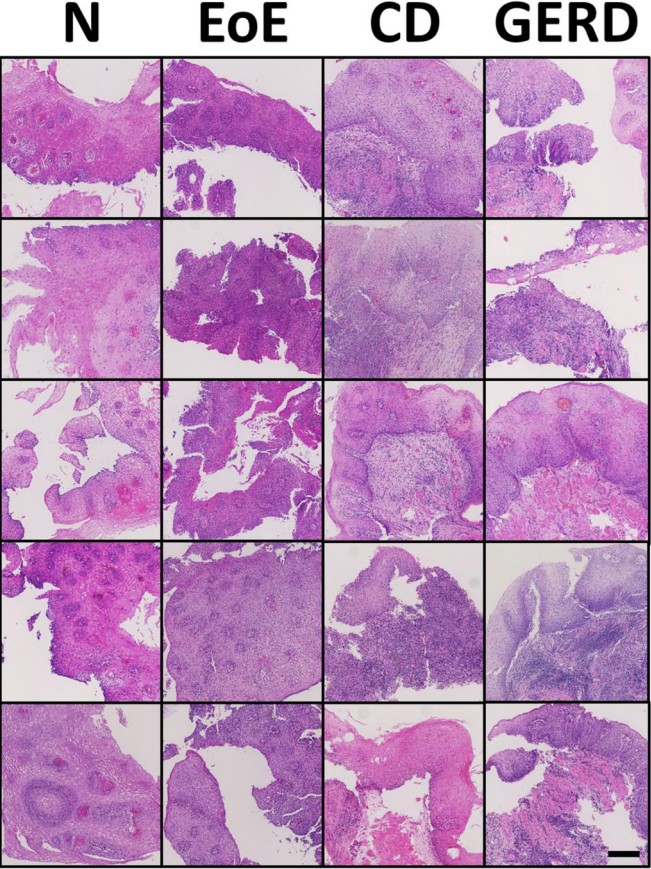

**Figure EV1. Histology of reference samples.**

Representative micrographs of H&E stains of the reference samples for Normal (*N*), Eosinophilic esophagitis (EoE), Crohn's disease (CD), and gastro-esophageal reflux disease (GERD). Scale bar: 200 µm.

**A**

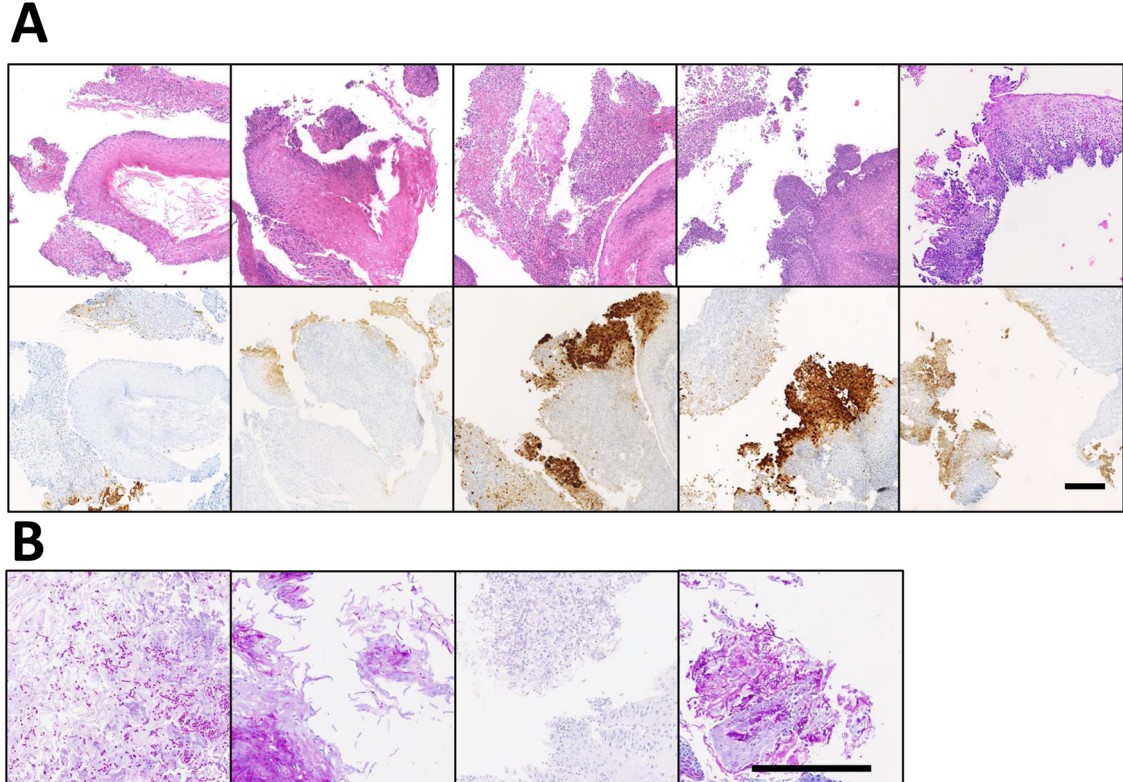

**B**

**Figure EV2. Histology of infectious samples.**

(A) H&E stains showing strong inflammation and heterogenous cytopathic changes (top row). IHC stains using an antibody against HSV1/2 highlight HSV-positive cells in brown (bottom row). (B) PAS reaction showing pseudo hyphae and spores of Candida spp. The epithelium displaying unspecific changes. Scale bars: 200 μm.

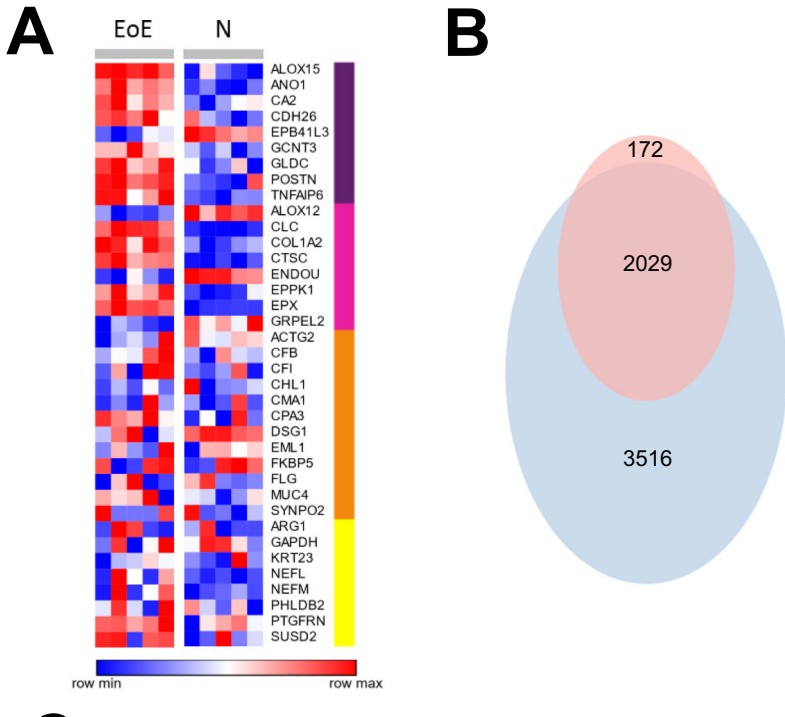

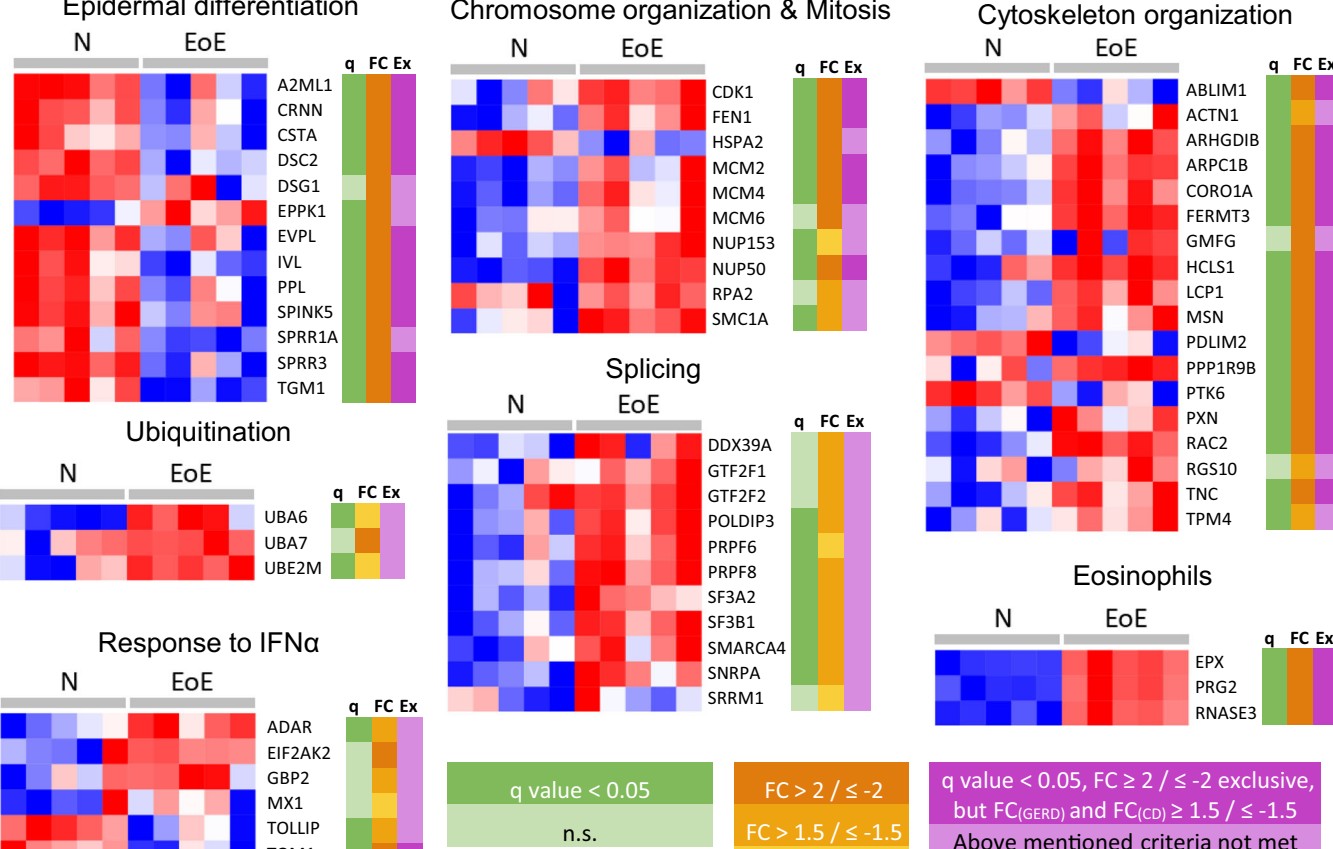

**Figure EV3. Subset of proteins quantified in EoE versus N corresponding to previously published markers for EoE.**

(A) Heatmap showing relative abundances of the indicated proteins in normal (*N*) and eosinophilic esophagitis (EoE) samples. The proteins correspond to markers (mRNAs) which were previously published (Wen et al, 2013) as part of an EoE diagnostic panel (EDP). Purple: exclusive for EoE v N; pink: significant, but not exclusive for EoE v N; orange: significant in CD v N or GERD v N, but not in EoE v N; yellow: not significantly altered, (B) Venn diagram of all quantified proteins in our dataset compared to a recent publication on tissue proteomics showing the differences between N and EoE samples (Rochman et al, 2023), (C) Heatmaps of the proteins mentioned in the enrichment analysis in Rochman et al, (2023) comparing N and EoE samples. Color coding showing different criteria for significance (q value, FC and exclusiveness in our dataset compared to GERD and CD samples) as stated on the bottom.

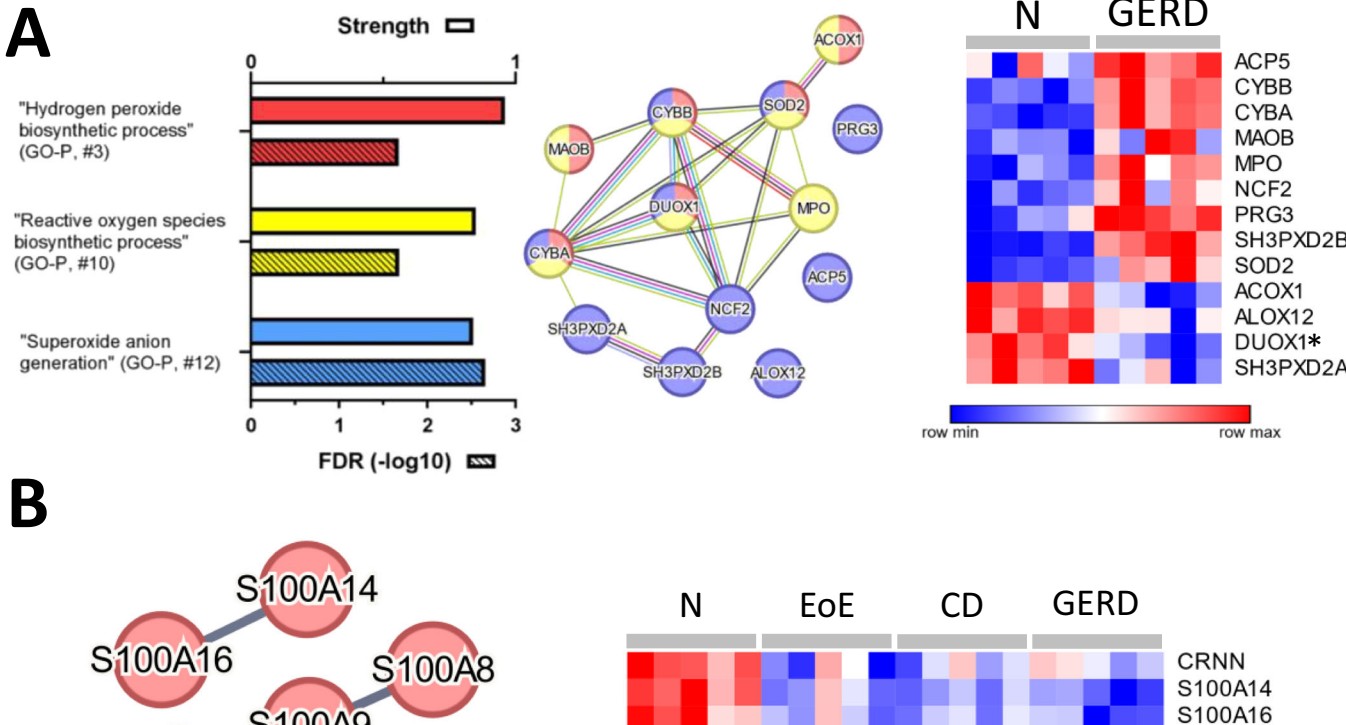

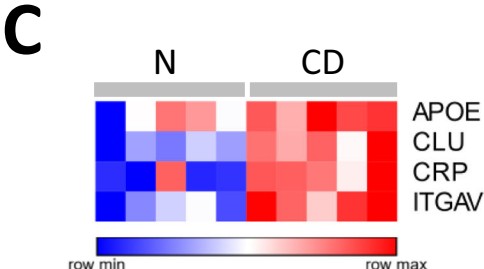

**Figure EV4. Other relevant proteins of different comparisons.**

(A) STRING networks of reactive oxygen species (ROS)-associated proteins and corresponding heatmaps (relative abundances of indicated proteins in each N and GERD sample) as well as related enrichment terms highlighted (for complete list see Dataset EV3B) by bar diagrams with the respective strength and false discovery rate (FDR, -log10, hatched bar) as well as the rank in the respective database. (B) STRING networks of S100A- and associated proteins and corresponding heatmaps (relative abundances of indicated proteins in each N, EoE, CD, and GERD sample). All of those proteins are significantly altered between N and each of the inflammatory groups. Except for S100A8 and S100A9 (each in N vs CD) all other proteins exceed log2FC 1/−1 (Dataset EV1C). (C) Heatmap (relative abundances of indicated proteins in each N and CD sample) of proteins previously described as predictive markers for anti-TNF-α therapy (Gazouli et al, 2013; Kalla et al, 2021; Kumar et al, 2024). The lowest value in each row in the heatmaps is displayed in dark blue, the highest value in dark red. The scale traverses white. The heatmaps were created using Morpheus.

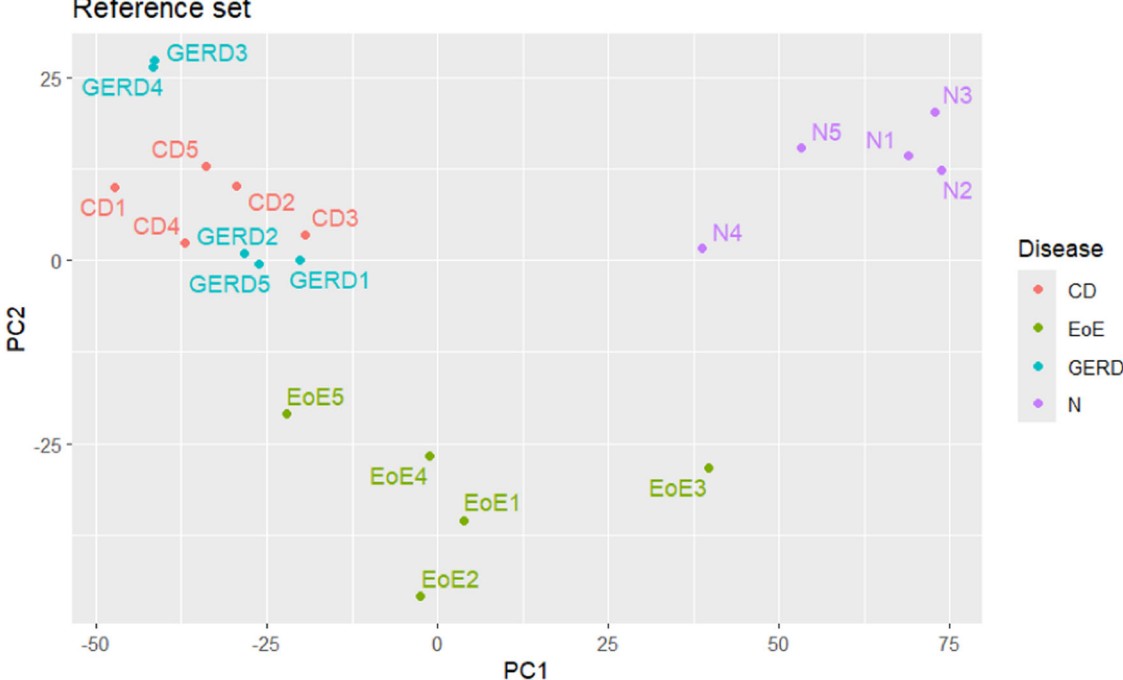

**Figure EV5. PCA plot of reference dataset.**

Principal component analysis (PCA) performed on the reference dataset illustrates separation of N, EoE, and the mixed cluster of GERD and CD samples.

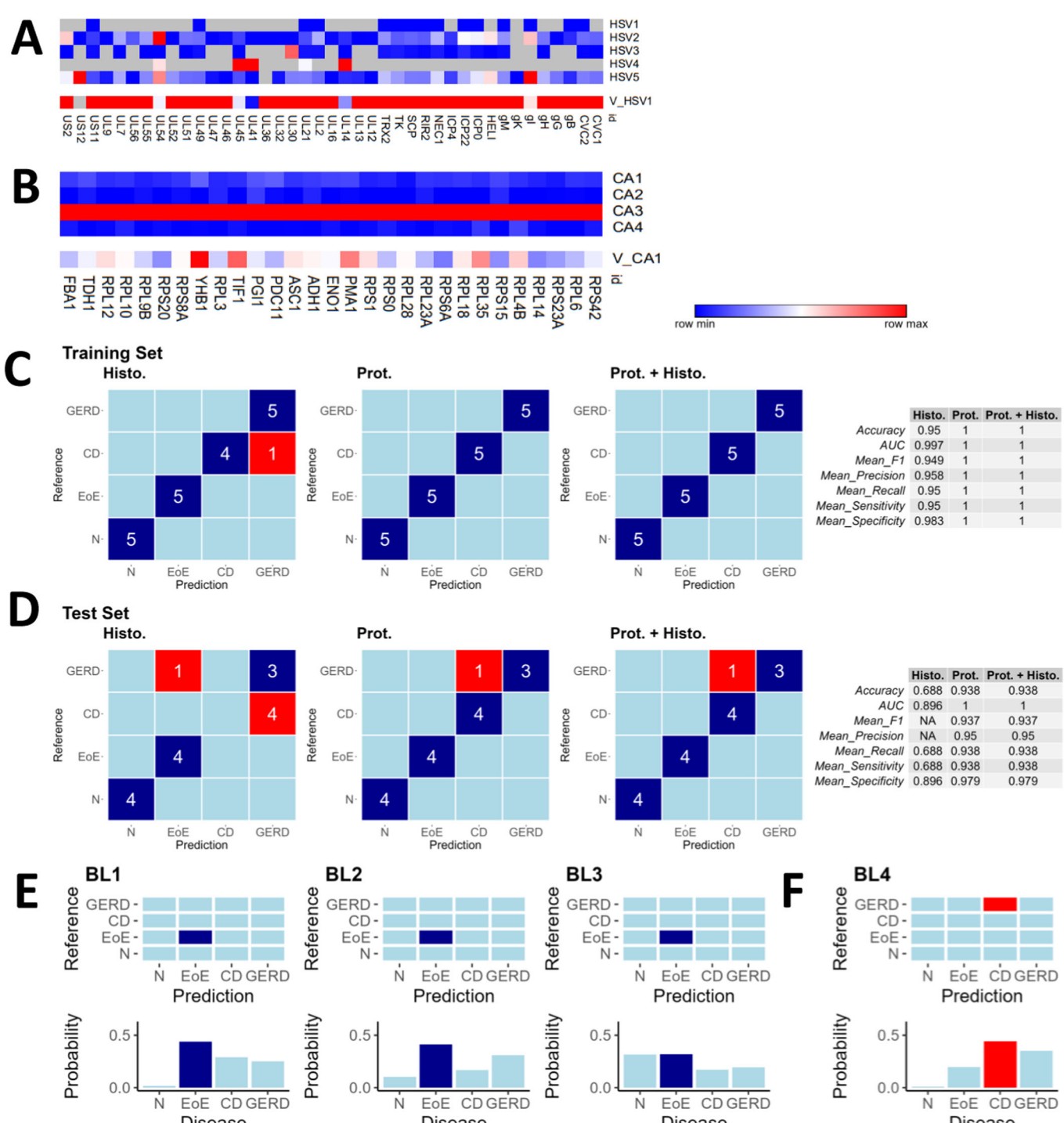

**Figure EV6. Validation of proteomic approach in infectious conditions, illustration of the random forest (RF) parameters in training and test samples, as well as results of borderline cases.**

(**A**) HSV-specific proteins are detected in the HSV esophagitis test sample (V_HSV1). (**B**) Candida-specific proteins are detected in the Candida esophagitis test sample (V_CA1). The lowest value in each column of the heatmaps is displayed in dark blue, the highest value in dark red. The scale traverses white. Heatmaps were created using Morpheus. (**C**) Results of the RF approach in the training dataset for histological features (Histo.) alone, proteome (Prot.) alone and combined (Prot. + Histo.). (**D**) Results of the RF with the aforementioned conditions in the test cohort. (**E**) Results of testing the approach in different borderline conditions. (**F**) Borderline case of a patient with histology and presentation favor GERD. Squares in the confusion matrices and bars colored in blue indicate the correct classification for each case by the RF, red markings indicating an incorrect classification.

