## [Peer Review File · EMBO Molecular Medicine]

An AI-assisted morphoproteomic approach is a supportive tool in esophagitis-related precision medicine

Sven Mattern, Vanessa Hollfoth, Eyyub Bag, Arslan Ali, Philip Riemenschneider, Mohamed Ali Jarboui, Karsten Boldt, Mihály Sulyok, Anabel Dickemann, Julia Luibrand, Stefano Fusco, Mirita Franz-Wachtel, Kerstin Singer, Benjamin Goeppert, Oliver Schilling, Nisar Malek, Falko Fend, Boris Macek, Marius Ueffing, and Stephan Singer

Corresponding author: Stephan Singer (stephan.singer@med.uni-tuebingen.de)

Review Timeline:

Submission Date:	3rd Nov 23
Editorial Decision:	20th Dec 23
Revision Received:	26th Jun 24
Editorial Decision:	2nd Sep 24
Revision Received:	4th Oct 24
Editorial Decision:	6th Dec 24
Revision Received:	14th Jan 25
Accepted:	14th Jan 25

Editor: Poonam Bheda

Transaction Report:

20th Dec 2023

Dear Dr. Singer,

Thank you for the submission of your manuscript to EMBO Molecular Medicine. We have now heard back from the referees who agreed to evaluate your manuscript. As you will see below, the reviewers raise substantial and overlapping concerns on your work. The main concerns to be addressed are:

- methods need to be described in full
- validation of proteomics targets needs to be included
- limitations of the study based on the small sample size should be discussed and conclusions toned down

If you feel you can satisfactorily address these points and those listed by the referees, you may wish to submit a revised version of your manuscript. A revised manuscript will once again be subject to review and we cannot guarantee at this stage that the eventual outcome will be favorable. If you would like to discuss further the points raised by the referees, I am available to do so via email or video.

We are expecting your revised manuscript within three months, if you anticipate any delay, please contact us.

We require:

- 1) A .docx formatted version of the manuscript text (including legends for main figures, EV figures and tables). Please make sure that the changes are highlighted to be clearly visible.
 - 2) Individual production quality figure files as .eps, .tif, .jpg (one file per figure). For guidance, download the 'Figure Guide PDF' (<https://www.embopress.org/page/journal/17574684/authorguide#figureformat>).
 - 3) At EMBO Press we ask authors to provide source data for the main figures. Our source data coordinator will contact you to discuss which figure panels we would need source data for and will also provide you with helpful tips on how to upload and organize the files.
 - 4) A .docx formatted letter INCLUDING the reviewers' reports and your detailed point-by-point responses to their comments. As part of the EMBO Press transparent editorial process, the point-by-point response is part of the Review Process File (RPF), which will be published alongside your paper.
 - 5) A complete author checklist, which you can download from our author guidelines (<https://www.embopress.org/page/journal/17574684/authorguide#submissionofrevisions>). Please insert information in the checklist that is also reflected in the manuscript. The completed author checklist will also be part of the RPF.
 - 6) Please note that all corresponding authors are required to supply an ORCID ID for their name upon submission of a revised manuscript.
 - 7) It is mandatory to include a 'Data Availability' section after the Materials and Methods. Before submitting your revision, primary datasets produced in this study need to be deposited in an appropriate public database, and the accession numbers and database listed under 'Data Availability'. Please remember to provide a reviewer password if the datasets are not yet public (see <https://www.embopress.org/page/journal/17574684/authorguide#dataavailability>).
- In case you have no data that requires deposition in a public database, please state so in this section. Note that the Data Availability Section is restricted to new primary data that are part of this study. This study includes no data deposited in external repositories.
- 8) For data quantification: please specify the name of the statistical test used to generate error bars and P values, the number (n) of independent experiments (specify technical or biological replicates) underlying each data point and the test used to calculate p-values in each figure legend. The figure legends should contain a basic description of n, P and the test applied. Graphs must include a description of the bars and the error bars (s.d., s.e.m.). Please provide exact p values.
 - 9) Our journal encourages inclusion of *data citations in the reference list* to directly cite datasets that were re-used and obtained from public databases. Data citations in the article text are distinct from normal bibliographical citations and should directly link to the database records from which the data can be accessed. In the main text, data citations are formatted as

follows: "Data ref: Smith et al, 2001" or "Data ref: NCBI Sequence Read Archive PRJNA342805, 2017". In the Reference list, data citations must be labeled with "[DATASET]". A data reference must provide the database name, accession number/identifiers and a resolvable link to the landing page from which the data can be accessed at the end of the reference. Further instructions are available at .

13) Author contributions: CRediT has replaced the traditional author contributions section because it offers a systematic machine readable author contributions format that allows for more effective research assessment. Please remove the Authors Contributions from the manuscript and use the free text boxes beneath each contributing author's name in our system to add specific details on the author's contribution. More information is available in our guide to authors.

Please also suggest a striking image or visual abstract to illustrate your article as a PNG file 550 px wide x 300-600 px high. Share synopsis text and image, as well as eTOC:

Please note that these would be the final versions and changes during proofing are usually not allowed

16) As part of the EMBO Publications transparent editorial process initiative (see our Editorial at <http://embomolmed.embopress.org/content/2/9/329>), EMBO Molecular Medicine will publish online a Review Process File (RPF) to accompany accepted manuscripts.

In the event of acceptance, this file will be published in conjunction with your paper and will include the anonymous referee reports, your point-by-point response and all pertinent correspondence relating to the manuscript. Let us know whether you agree with the publication of the RPF and as here, if you want to remove or not any figures from it prior to publication.

I look forward to receiving your revised manuscript.

Yours sincerely,

Poonam Bheda

Poonam Bheda, PhD
Scientific Editor
EMBO Molecular Medicine

**** Reviewer's comments ****

Referee #1 (Comments on Novelty/Model System for Author):

In this work, the authors performed a proteomic profiling of 40 biopsies from GERS, EOE CD, and HSV patients. The authors performed differential analyses to highlight important proteins and their interactions. The extracted proteome signatures were combined with histological parameters to train a random forest model to predict the esophagitis condition. Integrating molecular data into esophagitis diagnosis is important, yet there are some methodological concerns that have to be addressed.

1. The description of the statistical methods is lacking. In particular, it is unclear whether a multiple-hypothesis test was applied and which one. The authors should describe in detail their assumptions and the assumptions of the packages they use. This includes having numerical values for the various STRING outputs.
2. On the same note, the authors should clearly explain why they chose the particular STRING networks that are shown and explain that they were selected due to their statistical significance and were not cherry-picked.
3. The suggested relations between the results and their molecular mechanism should be explained better. For example, The relation of the results in Figure 2 to IL5, IL4, and IL13 should be supported with more direct explanations beyond the fact that they undergo alternative splicing.
4. The machine learning description is lacking. The authors should provide the confusion matrices and AUC for the training and test set (and the corresponding performance metrics such as accuracy, precision, recall, and F1-score). The authors should provide a detailed description of their data split procedure and show that their approach holds for different random splits.
5. The authors should quantify the marginal value of the proteomics features compared with histological parameters, that is, train separately and together and show quantified, significant improvement.

Referee #1 (Remarks for Author):

In this work, the authors performed a proteomic profiling of 40 biopsies from GERS, EOE CD, and HSV patients. The authors performed differential analyses to highlight important proteins and their interactions. The extracted proteome signatures were combined with histological parameters to train a random forest model to predict the esophagitis condition. Integrating molecular data into esophagitis diagnosis is important, yet there are some methodological concerns that have to be addressed.

1. The description of the statistical methods is lacking. In particular, it is unclear whether a multiple-hypothesis test was applied and which one. The authors should describe in detail their assumptions and the assumptions of the packages they use. This includes having numerical values for the various STRING outputs.
2. On the same note, the authors should clearly explain why they chose the particular STRING networks that are shown and explain that they were selected due to their statistical significance and were not cherry-picked.
3. The suggested relations between the results and their molecular mechanism should be explained better. For example, The relation of the results in Figure 2 to IL5, IL4, and IL13 should be supported with more direct explanations beyond the fact that they undergo alternative splicing.

4. The machine learning description is lacking. The authors should provide the confusion matrices and AUC for the training and test set (and the corresponding performance metrics such as accuracy, precision, recall, and F1-score). The authors should provide a detailed description of their data split procedure and show that their approach holds for different random splits.

5. The authors should quantify the marginal value of the proteomics features compared with histological parameters, that is, train separately and together and show quantified, significant improvement.

Referee #2 (Comments on Novelty/Model System for Author):

Detailed comments below

Referee #2 (Remarks for Author):

The proteomics methodology in the manuscript is not adequate. The described protocol does not mention key aspects of a proteomics study and the authors should provide more description for the manuscript to be considered for evaluation. The key aspects missing from the study are as follows -

1. It is not clear whether any control protein/peptides were spiked in the sample to account for run-to-run variation during the experiment.
2. The authors do not speak about the type of proteomics experiment in the study and should clarify whether the data was acquired using a DDA approach or a DIA approach. If the approach used is DDA, the authors must list out the reasons why this method was preferred over DIA which would have offered more comprehensive information.
3. There has been no effort made to provide and explain the overall experimental strategy. The abstract states that proteomic profiling was carried out for 40 biopsy specimens, but the 'statistics and proteomics signature conception' under the materials and methods section mentions the use of 20 samples for training. This adds to the ambiguity of the procedure followed in the study. The authors must clearly spell out how the data from 40 samples was used in the study. Additionally, were any samples excluded from the study? If yes, the authors must mention the reasons for excluding the samples from study.
4. Quantitative LC-MS approaches like MRM and PRM are used for validating targets from proteomics experiments. Did the authors carry out any such experiments to validate their targets? If not, the authors must include a statement clarifying the use of alternative validation approaches over MS-based validation approaches such as MRM and PRM.
5. The MS data analysis workflow is not explained in detail. For a manuscript that is this proteomics heavy, it is expected that all the details be mentioned at least in the supplementary section. The authors must explicitly state the pre-processing steps (normalization, imputation for missing values) used during analysis (if any).
6. Lastly, the claims made in the study seem a little far-fetched for a study containing 25 samples in the training set and 11 in the validation set.

Minor Comments -

1. The manuscript has typographical errors which can be easily resolved. The authors are requested to thoroughly proof-read the document prior to submission to avoid such mistakes. One such example is the typographical error with double commas in - „Biosynthesis of E-series 18(R)-resolvins“.
2. The PRIDE identifier ID is missing from the main text file. The authors are requested to provide this information for easy access of the uploaded data.

Referee #1 (Remarks for Author):

In this work, the authors performed a proteomic profiling of 40 biopsies from GERS, EOE CD, and HSV patients. The authors performed differential analyses to highlight important proteins and their interactions. The extracted proteome signatures were combined with histological parameters to train a random forest model to predict the esophagitis condition. Integrating molecular data into esophagitis diagnosis is important, yet there are some methodological concerns that have to be addressed.

1. The description of the statistical methods is lacking. In particular, it is unclear whether a multiple-hypothesis test was applied and which one. The authors should describe in detail their assumptions and the assumptions of the packages they use. This includes having numerical values for the various STRING outputs.

We totally agree with reviewer pointing out the necessity of multiple testing correction and apologize for the lack of detailed information in this regard. Perseus Software (Max-Planck Institute of Biochemistry) was employed for statistical testing. P-values were calculated via two-tailed student's t-test and adjusted by using permutation-based FDR resulting in respective q-values for which a < 0.05 significance level was defined. The details of the statistical methods are now provided in "Material and Methods" (*Data analyses and statistics*).

The detailed information/numerical values of all string outputs obtained in the respective comparisons ("term/pathway", "description", "count in network", "strength", and "FDR") are provided in Table EV2B, EV2F, EV3B/3C/3D, EV4B, EV5B/5C (see also response to comment "2.")

2. On the same note, the authors should clearly explain why they chose the particular STRING networks that are shown and explain that they were selected due to their statistical significance and were not cherry-picked.

We thank the reviewer for this important comment and would like to point out that all statistically significant STRING outputs according to the default settings of the String Database (<https://string-db.org/>): FDR < 0.05, strength \geq 0.01, and minimum count in network: 2) are part of our manuscript and are listed EV2B, EV2F, EV3B/3C/3D, EV4B, EV5B/5C. Unfortunately, we were not able to include all of these in the figures of the manuscript. The main criteria for presenting and discussing a particular network in the main or supplementary figures were a) that the network/term was among the top scoring hits (based on the strength value, at least top 20) and b) plausible in the given disease-specific context and c) could ideally be further supported by a focused literature search. For instance, based on the input "exclusively upregulated in CD esophagitis" results in the following KEGG pathway descriptions (top 5, based on the strength value): 1. "Allograft rejection" (none of our patients underwent transplantation) 2. "Primary immunodeficiency" (none of our patients suffers from a primary immunodeficiency disorder) 3. "Antigen processing and presentation" (is part of the pathophysiological concept of CD and therefore highlighted), 4. "Viral myocarditis" (none of the patients was diagnosed with viral myocarditis and esophageal biopsies are meaningless in this context), 5. "Graft versus host disease" (none of our patients underwent transplantation, as mentioned above). We assume that the reviewer would agree with us that the large majority of these terms appear not appropriate and are rather misleading within the given disease and tissue context, despite referring to partially overlapping protein groups. It may also reflect in this particular context a bias in the databases towards transplantation related research/publications and terms. With these limitations in mind we believe that the abovementioned way of rank-, context- and plausibility- based selection is acceptable, but surely is a compromise. Importantly, we explicitly state in "Materials and Methods" that the networks and terms highlighted in "Results and Discussion" and respective figures are by no means exhaustive and refer to the complete list of terms in the EV2B, EV2F, EV3B/3C/3D, EV4B, EV5B/5C, also in each corresponding figure legend.

3. The suggested relations between the results and their molecular mechanism should be explained better. For example, the relation of the results in Figure 2 to IL5, IL4, and IL13 should be supported with more direct explanations beyond the fact that they undergo alternative splicing.

We regret that the former version of the manuscript was apparently not clear enough in connecting the protein changes to relevant interleukin pathways. We have now included additional references, for instance Kouro et al. (2009) reporting that IL5 is responsible for recruitment of eosinophils in EoE, which we see at the morphological and proteomic level with the respective eosinophilic markers being highly abundant. In other words, the presence of eosinophils in the esophageal mucosa in EoE patients is the result of active IL5 signaling. Moreover, to further strengthen the link to IL4 and IL13 we performed a focused GSEA analyses to test if gene set members (in our case the corresponding proteins) of the IL4 and IL13 signaling pathway (Reactome Pathway Database) are overrepresented in the group of differentially abundant proteins in EoE patient samples. In fact, we observed a significant enrichment for gene/protein set members of the IL4 and IL13 pathway (see corresponding enrichment plot in new Figure 2D).

During the revision process we have conducted DIA analyses based on the suggestion of reviewer 2 with overall superior performance compared to DDA (presented before). However, for members of the splicing machinery it turns out that, while still being significantly altered based on their q values, they are no longer exclusive for EoE and largely do not exceed the log₂ FC 1 any more (see “reviewer only Figure 2”). Interestingly, a recently published paper by Rochman et al focusing on MCM in EoE (*JCI insight* (2023)) also reported overlapping splicing members to be overexpressed in their analyses. We refer to this finding in the EoE-related part of “Results and Discussion”.

4. The machine learning description is lacking. The authors should provide the confusion matrices and AUC for the training and test set (and the corresponding performance metrics such as accuracy, precision, recall, and F1-score). The authors should provide a detailed description of their data split procedure and show that their approach holds for different random splits.

We have now included a more detailed description of the machine learning approach. This includes information about the repeated cross-validation to determine the optimal parameters for our random forest classifier. Additionally, we provide confusion matrices and corresponding performance metrics as requested by the reviewer (AUC, accuracy, precision, recall, and F1-score) for the training dataset (n=20) and an additional test dataset (n=16) shown in Figure EV5C/D. Presenting now two separate datasets (training and test dataset) we did not randomly split a combined dataset.

5. The authors should quantify the marginal value of the proteomics features compared with histological parameters, that is, train separately and together and show quantified, significant improvement.

We thank the reviewer for this suggestion. We have fitted three different random forest classifiers. The first was only using the histological features, the second using only the proteome, and the third was the combination of both. In EV5C/D we now show how each of these classifiers performed on the training and test set. We illustrated these results with confusion matrices and performance metrics. The respective RF results show that proteome alone or proteome combined with histological features performed equally well compared to histology alone. However, we would not conclude that histology is negligible, because this comparable performance was observed by using histologically well evaluated samples. In general, no tissue-based molecular diagnostic analyses should be performed without prior histological (re-)evaluation of a given specimen regarding its composition and representativity. Moreover, in daily practice molecular analysis usually follows (not precedes) conventional histopathological diagnostics, which means that the histological information for each case is already available and integrable. Lastly, a subset of histological features are top ranked

variables with highest importance values in the combined RF approach (Table EV7) and may thereby contribute to a better performance of the combined RF compared to the “proteome only” RF in larger patient cohorts.

Referee #2 (Comments on Novelty/Model System for Author):

Detailed comments below

Referee #2 (Remarks for Author):

The proteomics methodology in the manuscript is not adequate. The described protocol does not mention key aspects of a proteomics study and the authors should provide more description for the manuscript to be considered for evaluation. The key aspects missing from the study are as follows –

1. It is not clear whether any control protein/peptides were spiked in the sample to account for run-to-run variation during the experiment.

Following the reviewer's suggestion (see point “2.”), the samples were re-measured at the proteomics core facility (CFMB in Tuebingen) using the DIA mode. To control for run-to-run variation, iRT standards were spiked into each sample, and their sequences were included in the DIA-NN search. We have included the report.pr_matrix.tsv and report.tsv files in our resubmission, which allow for checking the retention times (RT) of the iRT standard peptides and their quantification, indicating minimal run-to-run variability due to the LC-MS. Additionally, we have included the report.pdf from DIA-NN to demonstrate that all runs properly superpose (RT vs Library iRT), showing minimal run-to-run variability, which could primarily come from the LC. Finally, the consistent MS1/MS2 signal ratio across all samples further indicates minimal run-to-run variation during detection.

2. The authors do not speak about the type of proteomics experiment in the study and should clarify whether the data was acquired using a DDA approach or a DIA approach. If the approach used is DDA, the authors must list out the reasons why this method was preferred over DIA which would have offered more comprehensive information.

We thank the reviewer for this very important comment based on which we have re-analyzed all patient samples of the initial submission and included additional samples (n=15) by DIA. As anticipated by the reviewer, the DIA mode allowed strikingly better proteome coverage reflected by ~5500 quantified proteins (after filtering) compared to ~2800 of the initial DDA analysis. The DIA strategy is now mentioned in the abstract, in the “Results and Discussion” section as well as detailed in the “Material and Methods” section. We also provide “reviewer only” figures demonstrating side-by-side comparisons of the initial DDA protein signatures and the newly provided DIA signatures showing convincing correlations. Given that DIA clearly outperformed the previous DDA approach, we decided only to show the DIA acquired data.

3. There has been no effort made to provide and explain the overall experimental strategy. The abstract states that proteomic profiling was carried out for 40 biopsy specimens, but the 'statistics and proteomics signature conception' under the materials and methods section mentions the use of 20 samples for training. This adds to the ambiguity of the procedure followed in the study. The authors must clearly spell out how the data from 40 samples was used in the study. Additionally, were any samples excluded from the study? If yes, the authors must mention the reasons for excluding the samples from study.

We apologize for insufficient information about the overall experimental strategy. We now provide a respective paragraph in the “Results and Discussion” and a corresponding new Figure 1A explaining and illustrating, how the patient samples (n= 55 in total, including newly measured patient samples) are distributed in the respective groups.

The discovery/reference cohort consisted of 29 specimens (Fig. 1A) comprising control samples (n=5), eosinophilic esophagitis (EoE, n=5), esophageal Crohn’s disease (CD, n=5), gastro-esophageal reflux disease (GERD, n=5), Herpes simplex virus esophagitis (HSV, n=5) and Candida albicans esophagitis (CA, n=4).

The test cohort was composed of 18 samples (control (n=4), EoE (n=4), CD (n=4), GERD (n=4), HSV(n=1), and Candida (n=1)).

Finally, 8 cases were analyzed representing challenging and borderline (BL) patient samples as well as a CD patient discussed in the molecular inflammatory board (MIB).

The abovementioned discovery samples for EoE, GERD, CD, and normal controls served as a training set for the RandomForest (RF) (n=20). Validation samples (EoE (n=4), GERD (n=4), CD (n=4) and normal (n=4) served as validation set for the RandomForest (n=16).

All infectious esophagitis samples were not subjected to RF analyses as they were clearly identified by pathogen-specific proteins.

Patient samples were (in general) excluded from the study when their biopsy samples did not pass the quality controls in terms of tissue quality and representativity required for further analyses.

4. Quantitative LC-MS approaches like MRM and PRM are used for validating targets from proteomics experiments. Did the authors carry out any such experiments to validate their targets? If not, the authors must include a statement clarifying the use of alternative validation approaches over MS-based validation approaches such as MRM and PRM.

We believe that conducting DIA analyses on the entire patient cohort based on the reviewers' question (see point '2'), provided a form of validation, evidenced by the strong correlation of major EoE, GERD, and CD-related proteomic changes (see 'Reviewers only' figures). Additionally, we confirmed the proteomic alterations in EoE recently published by Rochman et al. (*JCI insight*, (2023), see new Figure EV3B/C and TableEV2D), derived from fresh frozen tissue, which can be considered as an external validation. These findings collectively support the robustness of our approach, even without conducting MRM or PRM for validating specific targets.

Furthermore, several studies suggest that DIA-MS methods are at least comparable to targeted methods. Boys et al. (2023) demonstrated the high accuracy of classifying histological subtypes and tissue-of-origin in tumor samples using a DIA-MS-based proteomic approach integrated with supervised machine learning. Nakamura et al. (2016) showed strong linear correlations in quantitative analysis between targeted and SWATH-MS results for human intestinal, liver, and kidney microsomes, providing comparable dynamic range and sensitivity. They concluded that, with consistent target tryptic peptides and sample preparation techniques, the choice between targeted and DIA-MS assays has minimal impact on LC-MS/MS-based protein quantification outcomes. More recently, Whelan et al. (2023) compared MRM-MS and DIA-MS-based assays for monitoring FDA-approved biomarkers in human plasma. DIA-MS offered expanded biomarker panel detection compared to MRM-based analysis, and both methods are deemed reliable for personal proteome biosignature stratification in precision medicine and health.

Therefore, we believe that DIA analysis can provide robust and reliable data comparable to most targeted methods. Nonetheless, we will consider targeted assays for future “pure” diagnostic studies.

5. The MS data analysis workflow is not explained in detail. For a manuscript that is this proteomics heavy, it is expected that all the details be mentioned at least in the supplementary section. The authors must explicitly state the pre-processing steps (normalization, imputation for missing values) used during analysis (if any).

The comprehensive MS data analysis, including parameters and data pre-processing steps, is now detailed in the “Materials and Methods” section and illustrated schematically in the new Figure 1C. Furthermore, in line with FAIR principles, parameters setting and log report files from our data analysis are as well submitted with the results files, so one can recreate the proteomics analysis by uploading the parameters settings and using DIA-NN (or Maxquant) versions reported in our Materials and Methods section. The outputs from DIA-NN LFQ protein intensities were already normalized at the peptide level, as we used maxLFQ algorithm integrated with DIA-NN and the cross-run normalization option based on RT to account for run-to-run variabilities, so no additional normalization was necessary. All samples were grouped according to their diagnosed disease and filtered for 60% values present in at least one group. Imputation was performed using normal distribution of 1.8 standard deviation down shift and with a width of 0.3 of the total data matrix (Perseus software, Max-Planck Institute of Biochemistry).

6. Lastly, the claims made in the study seem a little far-fetched for a study containing 25 samples in the training set and 11 in the validation set.

We agree with the reviewer that even with the expanded overall sample size of n=55 we should be more cautious in our conclusions. We therefore changed several phrases already starting with the title from “An AI-assisted morpho-proteomic approach is a *powerful* tool in esophagitis-related precision medicine” to “.....promising tool....” and corresponding changes were also made in the abstract and manuscript. We also used more phrases such as “suggest”, “may exhibit”, “may enhance”, “possibly involved in” etc. All edits to the manuscript are visible by the track changes mode.

Minor Comments –

1. The manuscript has typographical errors which can be easily resolved. The authors are requested to thoroughly proof-read the document prior to submission to avoid such mistakes. One such example is the typographical error with double commas in - „Biosynthesis of E-series 18(R)-resolvins”.

We apologize for these typographical errors, which we have corrected in the revised manuscript.

2. The PRIDE identifier ID is missing from the main text file. The authors are requested to provide this information for easy access of the uploaded data.

We uploaded the DIA-data and related relevant files on MassIVE, which can be accessed by

MassIVE ID: MSV000094966
ProteomeXchange: PXD052961

Reviewer-Account:
Nutzername: MSV000094966_reviewer
Passwort: UKTPathologie23!

Figures for reviewers removed

2nd Sep 2024

Dear Dr. Singer,

Thank you for the submission of your manuscript to EMBO Molecular Medicine. As discussed via email, the previous reviewers who evaluated your manuscript did not agree to re-review your paper, and therefore we procured two new reviewers to look over the previous reviewer concerns and your responses.

Both new reviewers were asked to give their opinion on whether the concerns of the previous reviewers had been addressed. While it seems that they do not raise any issues with the responses to the previous reviewers, you will see that they do raise concerns that seem critical, such as that the clinical impact is unclear and the methodology is not sufficiently described to promote reproducibility. While we would not necessarily ask you to address their concerns with new experiments, we do think that addressing some of their concerns would be helpful for the readers and for future reproducibility of your method and analyses. Therefore we would ask you to revise your manuscript with a full point-by-point response that will be sent back to these same reviewers.

If you feel you can satisfactorily address the points listed by the referees, please submit a revised version of your manuscript including a point-by-point response to their concerns. We can discuss this further during our scheduled phone call.

Please be reminded that we require the following:

4) A .docx formatted letter INCLUDING the reviewers' reports and your detailed point-by-point responses to their comments. As part of the EMBO Press transparent editorial process, the point-by-point response is part of the Review Process File (RPF), which will be published alongside your paper.

5) A complete author checklist, which you can download from our author guidelines (<https://www.embopress.org/page/journal/17574684/authorguide#submissionofrevisions>). Please insert information in the checklist that is also reflected in the manuscript. The completed author checklist will also be part of the RPF.

6) Please note that all corresponding authors are required to supply an ORCID ID for their name upon submission of a revised manuscript.

7) It is mandatory to include a 'Data Availability' section after the Materials and Methods. Before submitting your revision, primary datasets produced in this study need to be deposited in an appropriate public database, and the accession numbers and database listed under 'Data Availability'. Please remember to provide a reviewer password if the datasets are not yet public (see <https://www.embopress.org/page/journal/17574684/authorguide#dataavailability>).

In case you have no data that requires deposition in a public database, please state so in this section. Note that the Data Availability Section is restricted to new primary data that are part of this study. This study includes no data deposited in external repositories.

8) All Materials and Methods need to be described in the main text using our 'Structured Methods' format, which is required for all research articles. According to this format, the Methods section includes a Reagents and Tools Table (listing key reagents, experimental models, software and relevant equipment and including their sources and relevant identifiers) followed by a Methods and

Protocols section describing the methods using a step-by-step protocol format. The aim is to facilitate adoption of the methodologies across labs. Please upload the Reagents and Tools table as a separate document when submitting your revised manuscript. More information on how to adhere to this format as well as a downloadable template (.docx) for the Reagents and Tools Table can be found in our author guidelines:

<https://www.embopress.org/page/journal/17574684/authorguide#structuredmethods>

An example of a Method paper with Structured Methods can be found here:
<https://www.embopress.org/doi/10.15252/msb.20178071>.

9) For data quantification: please specify the name of the statistical test used to generate error bars and P values, the number (n) of independent experiments (specify technical or biological replicates) underlying each data point and the test used to calculate p-values in each figure legend. The figure legends should contain a basic description of n, P and the test applied. Graphs must include a description of the bars and the error bars (s.d., s.e.m.). Please provide exact p values.

10) Our journal encourages inclusion of *data citations in the reference list* to directly cite datasets that were re-used and obtained from public databases. Data citations in the article text are distinct from normal bibliographical citations and should directly link to the database records from which the data can be accessed. In the main text, data citations are formatted as follows: "Data ref: Smith et al, 2001" or "Data ref: NCBI Sequence Read Archive PRJNA342805, 2017". In the Reference list, data citations must be labeled with "[DATASET]". A data reference must provide the database name, accession number/identifiers and a resolvable link to the landing page from which the data can be accessed at the end of the reference. Further instructions are available at .

11) We replaced Supplementary Information with Expanded View (EV) Figures and Tables that are collapsible/expandable online. A maximum of 5 EV Figures can be typeset. EV Figures should be cited as 'Figure EV1, Figure EV2' etc... in the text and their respective legends should be included in the main text after the legends of regular figures.

12) The paper explained: EMBO Molecular Medicine articles are accompanied by a summary of the articles to emphasize the major findings in the paper and their medical implications for the non-specialist reader. Please provide a draft summary of your article highlighting

13) Author contributions: CRediT has replaced the traditional author contributions section because it offers a systematic machine readable author contributions format that allows for more effective research assessment. Please remove the Authors Contributions from the manuscript and use the free text boxes beneath each contributing author's name in our system to add specific details on the author's contribution. More information is available in our guide to authors.

Please also suggest a visual abstract to illustrate your article as a jpeg file 550 px wide x 300-600 px high.

Share synopsis text and image, as well as eTOC:

Please note that these would be the final versions and changes during proofing are usually not allowed

16) As part of the EMBO Publications transparent editorial process initiative (see our policy here:

https://www.embopress.org/transparent-process#Review_Process), EMBO Molecular Medicine will publish online a Peer Review File (PRF) to accompany accepted manuscripts.

In the event of acceptance, this file will be published in conjunction with your paper and will include the anonymous referee reports, your point-by-point response and all pertinent correspondence relating to the manuscript. Let us know whether you agree with the publication of the PRF and as here, if you want to remove or not any figures from it prior to publication.

I look forward to receiving your revised manuscript.

Yours sincerely,

Poonam Bheda

Poonam Bheda, PhD
Scientific Editor
EMBO Molecular Medicine

***** Reviewer's comments *****

Referee #3 (Comments on Novelty/Model System for Author):

To Editor: The authors seem to have addressed statistical and methodological concerns by Reviewers 1 and 2 who reviewed originally. This manuscript was hard to follow in their way of writing, but in part because I was given the "trackchanged and allmark up" version only.

In this revised manuscript, the authors performed LC-MS/MS-based proteomics upon formalin-fixed paraffin embedded (FFPE) tissues of endoscopic esophageal biopsies from patients with eosinophilic esophagitis (EoE), gastroesophageal reflux disease (GERD), Crohn's disease (CD), candida infection, and herpes simplex virus (HSV) infection. The authors demonstrate that the proteomics platform could detect disease relevant changes in the expression of many proteins. While some proteins are commonly expressed across differential disease conditions, pathogen-specific proteins were detected in candida and HSV-infected patient samples. CD samples displayed unique shift of the constitutive proteasome to the immunoproteasome. Finally, machine learning could accurately differentiate normal, EoE, GERD, and CD samples, validating the diagnostic value of this approach. This study confirmed the publicly available EoE transcriptomics and proteomics data. However, there are concerns as below.

Major points

1. Broadly focused, this study lacks critical controls such as esophageal samples from CD patients without esophageal lesions. EoE biopsies from the same patient may display variable histopathologic findings when taken from different sites in the esophagus. It is unclear to what extent, for example, the proteomics of EoE may quantitatively vary when biopsies are taken from the upper and lower esophagus from the same patient. It is unclear to what extent the authors' approach would be useful to understand the difference between EoE in disease remission (inactive EoE) and the active EoE cases featuring few eosinophilia (as highlighted in the Introduction section). Such cases were not included in this study. Clinical impact of this study is somewhat unconvincing.
2. What do the authors mean by "use cases" (Results and Discussion Section paragraph 1, Line 7)? What did these 8 cases serve in this study? If they were a part of experimental design, the rationale is not clearly written. I do not understand why they were included in the first paragraph of the Results section as "challenging and borderline" samples. The authors did not describe the rationale for the "test" cases (Fig. 1A) in the first paragraph, either. The workflow of research design and execution is difficult to follow.
3. At the end of paragraph 1, it is unclear whether a total of 5,554 protein groups represent "Discovery Approach" references only or all cases including the "test cohort" and the "user cases" (Fig. 1A). Additionally, it is unclear how various proteins were differentially expressed for each disease condition. Could principal component analysis (PCA) help to visualize variably expressed proteins amongst various diseases?
4. The authors state that data indicated both EoE and GERD samples had eosinophilia. It is unclear to what extent eosinophils-related proteins are differentially expressed between EoE and GERD. It would be surprising if most GERD showed eosinophils-related proteins to an extent that could not be distinguished from EoE. Additionally, it would be nicer if the authors compared the samples procured from the same EoE and GERD patients before and after proton pump inhibitors (PPIs) therapy which is used

for both disease conditions.

5. While machine learning was able to distinguish EoE from GERD and other conditions, it is unclear what factors, either histologic findings or proteins (e.g., epithelial cell proliferation/differentiation status, immune cell-related molecules, stromal fibroblasts), made it possible at the end. For example, it is not clear to me about what makes EoE unique even in the absence of eosinophilia and how machine learning could distinguish EoE from GERD.

6. No validation (e.g., immunohistochemistry) was offered for the proteins that were linked to any disease conditions.

Minor points

The end of the second paragraph (2nd line from the bottom): There is a typo. It must be "function", the word following antagonistic.

Referee #3 (Remarks for Author):

In this revised manuscript, the authors performed LC-MS/MS-based proteomics upon formalin-fixed paraffin embedded (FFPE) tissues of endoscopic esophageal biopsies from patients with eosinophilic esophagitis (EoE), gastroesophageal reflux disease (GERD), Crohn's disease (CD), candida infection, and herpes simplex virus (HSV) infection. The authors demonstrate that the proteomics platform could detect disease relevant changes in the expression of many proteins. While some proteins are commonly expressed across differential disease conditions, pathogen-specific proteins were detected in candida and HSV-infected patient samples. CD samples displayed unique shift of the constitutive proteasome to the immunoproteasome. Finally, machine learning could accurately differentiate normal, EoE, GERD, and CD samples, validating the diagnostic value of this approach. This study confirmed the publicly available EoE transcriptomics and proteomics data. However, there are concerns as below.

Major points

1. Broadly focused, this study lacks critical controls such as esophageal samples from CD patients without esophageal lesions. EoE biopsies from the same patient may display variable histopathologic findings when taken from different sites in the esophagus. It is unclear to what extent, for example, the proteomics of EoE may quantitatively vary when biopsies are taken from the upper and lower esophagus from the same patient. It is unclear to what extent the authors' approach would be useful to understand the difference between EoE in disease remission (inactive EoE) and the active EoE cases featuring few eosinophilia (as highlighted in the Introduction section). Such cases were not included in this study. Clinical impact of this study is somewhat unconvincing.

2. What do the authors mean by "use cases" (Results and Discussion Section paragraph 1, Line 7)? What did these 8 cases serve in this study? If they were a part of experimental design, the rationale is not clearly written. I do not understand why they were included in the first paragraph of the Results section as "challenging and borderline" samples. The authors did not describe the rationale for the "test" cases (Fig. 1A) in the first paragraph, either. The workflow of research design and execution is difficult to follow.

3. At the end of paragraph 1, it is unclear whether a total of 5,554 protein groups represent "Discovery Approach" references only or all cases including the "test cohort" and the "user cases" (Fig. 1A). Additionally, it is unclear how various proteins were differentially expressed for each disease condition. Could principal component analysis (PCA) help to visualize variably expressed proteins amongst various diseases?

4. The authors state that data indicated both EoE and GERD samples had eosinophilia. It is unclear to what extent eosinophils-related proteins are differentially expressed between EoE and GERD. It would be surprising if most GERD showed eosinophils-related proteins to an extent that could not be distinguished from EoE. Additionally, it would be nicer if the authors compared the samples procured from the same EoE and GERD patients before and after proton pump inhibitors (PPIs) therapy which is used for both disease conditions.

5. While machine learning was able to distinguish EoE from GERD and other conditions, it is unclear what factors, either histologic findings or proteins (e.g., epithelial cell proliferation/differentiation status, immune cell-related molecules, stromal fibroblasts), made it possible at the end. For example, it is not clear to me about what makes EoE unique even in the absence of eosinophilia and how machine learning could distinguish EoE from GERD.

6. No validation (e.g., immunohistochemistry) was offered for the proteins that were linked to any disease conditions.

Minor points

The end of the second paragraph (2nd line from the bottom): There is a typo. It must be "function", the word following

antagonistic.

Referee #4 (Remarks for Author):

More detail to follow, but the first question I had when reading this was if I could repeat this work with the methods provided. I do not think so. Some of the details in the Material and Methods are in complete, or parts are incorrect, I think.

Further, from the previous reviewers comments it seems there was a lack of information, as both reviewers had complaints, albeit different complaints. I would have had questions related to the presented methods as well. I still have questions about some of the information on the proteomics methods reported. Like the others, I am also concerned about the hype here as we read in the new title still that: "An AI-assisted morphoproteomic approach is a promising tool in esophagitis-related precision medicine.". While I understand the sentiment, I would expect that at least 10 times the number of experiments compared to proteins would be used for better learning and ML application. I cannot comment for the rest on the ML work or statistics used.

I read the manuscript with improvements, but there seems to be a mistake in the number of patients, and how this number is discussed. In the abstract there is mention of 55 samples, yet in the Materials and Methods I count 56 samples.

Here some other observations / thoughts:

Patient cohort and samples

The collection of FFPE-tissue samples used in this study consisted exclusively of endoscopic esophageal biopsies obtained for histopathological routine diagnostics between 2015 and 2023. Patient samples were excluded from the study when their biopsy samples did not pass the quality controls in terms of tissue quality and representativity required for further analyses. All Samples (total n=55; Fig. 1A) were retrieved from the archive of the Institute of Pathology of the University Hospital Tuebingen, Germany. The cohort for the exploratory approach contained different types of esophagitis: GERD (n=5), CD (n=5), EoE (n=5) and HSV (n=5) as well as control samples showing no overt inflammation or other alterations (Normal, N, n=5). This cohort (n=20 is this not 25??) also served as a training set for the machine learning approach (Random Forest (RF), see below). For test of the RF another cohort was compiled consisting of 16 samples (GERD (n=4), CD (n=4), EoE (n=4), N (n=4)). Moreover, the RF was tested using challenging or borderline cases. Every sample was evaluated by experienced pathologists (S.S., K.S., S.M., F.F.). **For validation/ clarification 15 additional** samples including cases with ambiguous/unspecific histological findings were analysed. Patient characteristics are summarized in Table EV1A.*

Sample size: 25+16+15 = 56

Liquid chromatography-Mass Spectrometry (LC-MS/MS)

Q1

Peptides were eluted and separated on the PharmaFluidics μ PAC nano-LC column: 50 cm μ PAC C18 with a pillar diameter of 5 μ m, inter-pillar distance of 2.5 μ m, pillar length/bed depth of 18 μ m, external porosity of 59%, bed channel width of 315 μ m and bed length of 50 cm; pillars are superficially porous with a porous shell thickness of 300 nm and pore sizes in the order of 100 to 200 Å by a linear gradient from 2% to 30 % of buffer B (80% acetonitrile and 0.08% formic acid in HPLC-grade water) in buffer A (2% acetonitrile and 0.1% formic acid in HPLC-grade water) at a flow rate of 300 nl per min. The remaining peptides were eluted by a short gradient of 10 minutes from 30% to 95% buffer B; followed by 25 minutes at 2% of buffer B, the total gradient run was 120 min.

FOR HOW LONG was the LC gradient developed? (120-10-25=85 min?)**

Q2

There is no explanation of the "two BSA washes were used between samples". Please provide a reference or details (e.g. 1% ??) , as not all researchers know this 'trick'.

Data analysis

Q3

*...the 11 iRT peptide sequences were manually added to **the database** and used during DIA-NN search to generate the precursor ion library used for MS data analysis.*

For a reference to the database please add see *Data analysis and statistics*

*...the 11 iRT peptide sequences were manually added to **the database**(see below Data analysis and statistics) and used during DIA-NN search to generate the precursor ion library used for MS data analysis. *

Q4

LC-MS/MS was performed at the Proteome Center Tuebingen (PCT). Here, 250 ng of peptides were loaded onto an Easy-nLC 1200 system coupled to a quadrupole Orbitrap Exploris 480 mass spectrometer (all Thermo Fisher Scientific, Waltham, MA, USA) as previously described (Krauss et al, 2023).

Shouldn't reference of Kraus be omitted? *(Krauss et al, 2023).*

Q5

It is not clear to me why iRTs used when DIA-NN was used (in library free mode) as a search engine?

I anticipate others (e.g. Rochman group) are keen to compare this work with their own, so would encourage all steps to be described precisely.

Q6

I do not understand the reporting here. Some of the text seems superfluous:

The article states that DIA-NN v1.8.1 is used and then go about explaining the steps the program performs, whereas these steps are merely 'clickable' selections in the DIA-NN GUI. The way it is written here, it seems like these are independent steps by the user. For example: DIA-NN allows the choice of one's own library, or a FASTA formatted library (from PRIDE or SwissProt, e.g.). The authors state:

"First, a precursor ion library was generated using FASTA digest for library-free search in combination with deep learning- based spectra prediction. An experimental library generated from the DIA-NN search was used for cross-run normalization and mass accuracy correction. Only high-accuracy spectra with a minimum precursor FDR of 0.01, and only tryptic peptides (2 missed Tryptic cleavages) were used for protein quantification. "

We highly appreciate that Reviewer #3 and Reviewer #4 stepped in during the review process, given that Reviewer #1 and Reviewer #2 unfortunately declined to re-evaluate our revised manuscript. As explained to us by the editor, the primary request to Reviewer #3 and Reviewer #4 was to assess whether we had adequately addressed the comments and concerns raised by Reviewer #1 and Reviewer #2. While we believe that several of the current comments and suggestions extend beyond this aforementioned review request we nevertheless have carefully considered them and provided detailed responses in an effort to further improve the quality of our manuscript.

Reviewer comments:

Reviewer #3

In this revised manuscript, the authors performed LC-MS/MS-based proteomics upon formalin-fixed paraffin embedded (FFPE) tissues of endoscopic esophageal biopsies from patients with eosinophilic esophagitis (EoE), gastroesophageal reflux disease (GERD), Crohn's disease (CD), candida infection, and herpes simplex virus (HSV) infection. The authors demonstrate that the proteomics platform could detect disease relevant changes in the expression of many proteins. While some proteins are commonly expressed across differential disease conditions, pathogen-specific proteins were detected in candida and HSV-infected patient samples. CD samples displayed unique shift of the constitutive proteasome to the immunoproteasome. Finally, machine learning could accurately differentiate normal, EoE, GERD, and CD samples, validating the diagnostic value of this approach. This study confirmed the publicly available EoE transcriptomics and proteomics data. However, there are concerns as below.

Major points

1. Broadly focused, this study lacks critical controls such as esophageal samples from CD patients without esophageal lesions. EoE biopsies from the same patient may display variable histopathologic findings when taken from different sites in the esophagus. It is unclear to what extent, for example, the proteomics of EoE may quantitatively vary when biopsies are taken from the upper and lower esophagus from the same patient. It is unclear to what extent the authors' approach would be useful to understand the difference between EoE in disease remission (inactive EoE) and the active EoE cases featuring few eosinophilia (as highlighted in the Introduction section). Such cases were not included in this study. Clinical impact of this study is somewhat unconvincing.

We totally agree with the reviewer that esophageal samples from CD patients without esophageal lesions are important and that is why these were already included in the study along with a sample of a CD patient suffering from GERD. As shown in Figure 7C and written in the Results and Discussion section: "To further challenge the morpho-proteomic RF we used inconspicuous esophageal samples of 2 CD patients (N1(CD) and N2(CD)) and a case of a patient with intestinal CD, but with the histological diagnose of GERD (GERD(CD)). As shown in Figure 7C each case was correctly predicted as N or GERD, respectively."

We also agree that there could be localization-dependent proteomic alterations in the esophagus under physiological and pathophysiological conditions. That is exactly the reason why we specifically included only samples of the lower esophagus (which represents the most common site of biopsy collection) ensuring that disease-related changes were not confounded by potential topographical proteomic variations.

The main focus of the manuscript was, indeed, on patient samples with fully developed diseases (EoE, GERD, and CD), rather than on the corresponding inactive states of each disease. We believe that comparing the inactive and active manifestations for each condition would exceed the scope of the current study, but represents a promising direction for future research. Additionally, a truly inactive state of a disease is expected to be largely symptomless and endoscopically inconspicuous so

that (strictly speaking) there would be no indication for a biopsy (see also comment 4.). However, our study does consider the challenging diagnostic situation of patients with a clinical diagnosis of EoE, but with eosinophil counts below the current diagnostic threshold (<15/HPF, paucicellular EoE), as well as a biopsy from an EoE patient, the appearance of which was nearly normal (approaching an inactive state).

As stated in the Results and Discussion: “Moreover, we also tested borderline (BL) cases of EoE patients (BL1-3) and one borderline GERD case (BL4) with the following characteristics and results. BL1 was histologically characterized by only very few foci of >15 Eos/HPF as well as large areas without significant eosinophilic infiltrates and BL2 showed an increased number of Eos/HPF, however, not exceeding the cut off >15/HPF in any area. Both were predicted as EoE, which we consider as the correct diagnosis in the given context (Fig. EV5E, left and middle panel). BL3 presented a rather normal histological appearance with only very few eosinophils (2/HPF) in the biopsy being molecularly analyzed (in contrast to another biopsy submitted at the same time with clear cut EoE histology not subjected to proteomic analyses). For BL3 the RF favoured EoE, however, with an almost equal predictive value for N (Fig. EV5E, right panel) together with the aforementioned BL cases suggesting this approach as a useful tool in pauci-cellular EoE.”

2. What do the authors mean by "use cases" (Results and Discussion Section paragraph 1, Line 7)? What did these 8 cases serve in this study? If they were a part of experimental design, the rationale is not clearly written. I do not understand why they were included in the first paragraph of the Results section as "challenging and borderline" samples. The authors did not describe the rationale for the "test" cases (Fig. 1A) in the first paragraph, either. The workflow of research design and execution is difficult to follow.

We regret that the term “use cases” was apparently not clearly explained. Our intention was to use the term to refer to application examples of the morpho-proteomic RF in challenging or borderline diagnostic cases. In contrast, the training and test cohorts comprised more typical manifestations of each disease. In other words, the training set was used to train the RF on relatively clear-cut cases and the test cohort to evaluate the RF’s performance on similarly well-defined samples. Finally, the “use cases” were included to evaluate the supportiveness of the RF in “real-world” challenging/borderline diagnostic settings. We have added additional explanatory sentences in the Material and Methods section and respective paragraph of the Results and Discussion section.

3. At the end of paragraph 1, it is unclear whether a total of 5,554 protein groups represent "Discovery Approach" references only or all cases including the "test cohort" and the "user cases" (Fig. 1A). Additionally, it is unclear how various proteins were differentially expressed for each disease condition. Could principal component analysis (PCA) help to visualize variably expressed proteins amongst various diseases?

The 5,554 protein groups refer to all analysed samples including the test subcohort and use cases. We have clarified this accordingly (see first paragraph in Results & Discussion).

For each disease (Figures 2-4, Figure EV3, EV4, Tables EV2A, EV3A, EV4A), we present both overall differentially abundant proteins (DAPs) and exclusive DAPs. The exclusive DAPs are of particular interest as they indicate disease-specific proteomic alterations. This means that exclusive DAPs are not altered in the same direction according to our defined q- and log2FC-thresholds in the other N vs disease comparisons. We actually do believe that this allows quite detailed insights about differentially expressed proteins for each disease condition, which is further complemented by DAPs

that are shared across N vs disease comparisons (Figure 5, Table EV5A). Any additional analyses and comparisons beyond the current data presentation including provided volcano plots, heatmaps, and tables can be extracted from the complete data files deposited on MassIVE.

In our understanding, PCA is primarily performed for dimensional reduction of high-dimensional data, allowing it to be visualized and helping to identify clusters of samples or groups (in our case, disease conditions). Based on the reviewers' suggestion, we conducted a principal component analysis (PCA), which resulted in the following PCA plot for the reference samples.

Figure 1: PCA-Plot illustrating the similarity/separation of the reference samples.

The PCA plot indicates that the highest similarity exists between the GERD and CD cases, while the EoE and N samples cluster separately.

Figure 2: Percentage contributions of the 30 most contributing proteins to the first (PC1) and second principal component (PC2).

Additionally, we employed PCA to quantify the individual contributions of the quantified proteins to the first and second principal components (PC1 and PC2, respectively). Figure 2 highlights the contribution values for the Top 30 contributing proteins, revealing that these values do not strikingly differ between variables. This indicates that many proteins demonstrate a high degree of co-expression - and consequently appear as co-linear features sharing the variance in the PCA. We believe that the volcano plots, heatmaps, and tables provided in the current version of the manuscript are a better representation of the expression changes driving group differences.

4. The authors state that data indicated both EoE and GERD samples had eosinophilia. It is unclear to what extent eosinophils-related proteins are differentially expressed between EoE and GERD. It would be surprising if most GERD showed eosinophils-related proteins to an extent that could not be distinguished from EoE. Additionally, it would be nicer if the authors compared the samples procured from the same EoE and GERD patients before and after proton pump inhibitors (PPIs) therapy which is used for both disease conditions.

As mentioned above, the primary focus of the study was to identify exclusive DAPs for each disease condition compared to normal tissue (besides overall significant DAPs). Eosinophilic granulocyte markers were significantly upregulated in all disease conditions (EoE, GERD, and also CD) and thus did not fall into the group of exclusive DAPs. However, as the reviewer correctly pointed out, there should be higher levels of eosinophilic markers in EoE compared to GERD. This difference can indeed be seen when comparing Figure 2A and Figure 3A. In the respective volcano plots, the log₂ FC of eosinophil-related proteins, such as EPX, RNASE2, and RNASE, is ~ 6 in GERD and ~ 9 in EoE.

According to the German S2k guideline for the diagnosis and treatment of gastroesophageal reflux disease (GERD) (as of 2022), the administration of proton pump inhibitors (PPIs) is based on the patient's symptoms and does not require a prior biopsy. In fact, a biopsy is only recommended if symptoms persist despite PPI therapy or if there is an unclear endoscopic aspect. Conducting a biopsy before starting PPIs would not be in line with these guidelines. While the comparison before and after PPI treatment could be of potential interest it would require a prospective study and thus goes beyond the scope and focus of the current investigation.

5. While machine learning was able to distinguish EoE from GERD and other conditions, it is unclear what factors, either histologic findings or proteins (e.g., epithelial cell proliferation/differentiation status, immune cell-related molecules, stromal fibroblasts), made it possible at the end. For example, it is not clear to me about what makes EoE unique even in the absence of eosinophilia and how machine learning could distinguish EoE from GERD.

To address the reviewers' question, we refer to the supplementary table (EV Table 7). This table lists the weighted importance values from the Random Forest machine learning approach (Importance Prot. + Histo.). These values reflect how much each feature contributes to improving the predictions, indicating a relative measure of each feature's impact on the model's decisions. The top-ranked histological features include "focus of the eosinophilic infiltrate" and "inflammation focus," or "severity of inflammation". Proteomic features include eosinophilic markers such as RNASE3 and CLC, as well as the neutrophilic marker CTSC, alongside many other proteins. These weighted importance values pertain to the differentiation between all disease groups, not just between EoE and GERD. Using only the proteins included in the TOP25 variables (Figure 3) and excluding the histology features already produces a reasonably good, albeit not perfect, clustering and separation of the disease groups (particularly of EoE and GERD samples, see Figure 4). While this is not representing the comprehensive results of the RF, it may help to get an impression.

Figure 3: TOP25 variables of the histo-proteomic RF according to their weighted importance values

Figure 4 Heatmap and hierarchical clustering (Parameters in R: distance: euclidean, method: complete) of reference, validation, and “use cases” (borderline and challenging samples) just using protein variables of the TOP25 variables with highest weighted importance values of the RF.

Random Forest classifiers classify a sample by using the most voted classification across a large number of decision trees (1,000 in our study). Each tree considers only a subset of features and

samples for training, which helps reduce variance and enhance overall performance. Given that the classification of an individual sample is the result of 1,000 trees, it becomes extremely difficult to pinpoint the exact factors used for classifying that particular sample (e.g. EoE with few or almost no eosinophils, as mentioned by the reviewer).

6. No validation (e.g., immunohistochemistry) was offered for the proteins that were linked to any disease conditions.

As demonstrated in the manuscript and already noted by the reviewer, we were able to reproduce a significant portion of previously published proteomic and transcriptomic data on EoE. We consider this to be a substantial validation. Additionally, concordant differentially expressed proteins included those previously demonstrated by immunohistochemistry to be altered in EoE, such as strong ALOX15 positivity in EoE biopsy samples (e.g. Hui Y et al. (2017). *Pediatr Dev Pathol* 20: 375-380, which is included in our references). Furthermore, we observed a tight correlation of the proteomic changes with histomorphological findings (immune infiltrate composition, epithelial alterations, fibrosis, etc). This together reinforces our confidence in the validity of our approach and we are not convinced that selecting additional markers for further immunohistochemical analyses would significantly improve the manuscript.

Minor points

The end of the second paragraph (2nd line from the bottom): There is a typo. It must be "function", the word following antagonistic.

Thank you for pointing this out, we have corrected this typographical error.

Reviewer #4

More detail to follow, but the first question I had when reading this was if I could repeat this work with the methods provided. I do not think so. Some of the details in the Material and Methods are incomplete, or parts are incorrect, I think.

Further, from the previous reviewers comments it seems there was a lack of information, as both reviewers had complaints, albeit different complaints. I would have had questions related to the presented methods as well. I still have questions about some of the information on the proteomics methods reported. Like the others, I am also concerned about the hype here as we read in the new title still that: "An AI-assisted morphoproteomic approach is a promising tool in esophagitis-related precision medicine.". While I understand the sentiment, I would expect that at least 10 times the number of experiments compared to proteins would be used for better learning and ML application. I cannot comment for the rest on the ML work or statistics used.

We regret that even after revising the title from ‘...powerful tool...’ to ‘...promising tool..’ the reviewer still considers the title inappropriate. We have now further adjusted the title to ‘An AI-assisted morphoproteomic approach is a supportive tool in esophagitis-related precision medicine.’ In full agreement with the reviewer we also think that increasing the number of experiments/samples is beneficial for developing, training, and testing machine learning approaches. However, if the reviewer is indeed referring to samples as ‘experiments’, ‘at least 10 times the number of experiments compared to proteins’ would translate to $10 \times 5500 = 55\,000$ (!) samples in our case, which seems impractical. We believe that this is most likely a misunderstanding on our part.

The clarifications of the Material and Methods related comments are responded to below.

I read the manuscript with improvements, but there seems to be a mistake in the number of patients, and how this number is discussed. In the abstract there is mention of 55 samples, yet in the Materials and Methods I count 56 samples.

We are grateful for the acknowledgement of the improvements made to the manuscript based on the suggestions of Reviewer #1 and #2. The number of 55 samples is correct, as explained below.

Here some other observations / thoughts:

Patient cohort and samples

The collection of FFPE-tissue samples used in this study consisted exclusively of endoscopic esophageal biopsies obtained for histopathological routine diagnostics between 2015 and 2023. Patient samples were excluded from the study when their biopsy samples did not pass the quality controls in terms of tissue quality and representativity required for further analyses.

*All Samples (total n=55; Fig. 1A) were retrieved from the archive of the Institute of Pathology of the University Hospital Tuebingen, Germany. The cohort for the exploratory approach contained different types of esophagitis: GERD (n=5), CD (n=5), EoE (n=5) and HSV (n=5) as well as control samples showing no overt inflammation or other alterations (Normal, N, n=5). This cohort (n=20 is this not 25??) also served as a training set for the machine learning approach (Random Forest (RF), see below). For test of the RF another cohort was compiled consisting of 16 samples (GERD (n=4), CD (n=4), EoE (n=4), N (n=4)). Moreover, the RF was tested using challenging or borderline cases. Every sample was evaluated by experienced pathologists (S.S., K.S., S.M., F.F.). **For validation/clarification 15 additional** samples including cases with ambiguous/unspecific histological findings were analysed. Patient characteristics are summarized in Table EV1A.**

Sample size: 25+16+15 = 56

We apologize for any confusion regarding the total number of samples analyzed in the study (n=55), which may have been partly caused by the extensive markups and tracked changes in the revised manuscript. In the “clean version” of the re-revised manuscript the respective paragraph in the Material and Methods now states: “All Samples (total n=55; Fig. 1A) were retrieved from the archive of the Institute of Pathology of the University Hospital Tuebingen, Germany. The subcohort for the exploratory approach contained different types of esophagitis: GERD (n=5), CD (n=5), EoE (n=5) as well as control samples showing no overt inflammation or other alterations (Normal, N, n=5). This subcohort (n=20) also served as a training set for the machine learning approach (“Training Set (RF)”). For testing the performance of the RF another subcohort (“Test Set (RF)”) was compiled consisting of 16 samples (GERD (n=4), CD (n=4), EoE (n=4), N (n=4)). Moreover, the RF was applied to challenging or borderline cases (“use cases” (n = 8): borderline EoE (BL1-3, n=3), borderline GERD (BL4, n=1), CD patients with normal esophageal biopsies (N(CD), n=2), a CD patient with GERD (N(GERD), n=1) and a CD patient discussed in the molecular inflammation board with the external diagnosis GERD (MIB, n=1)). The infectious esophagitis samples comprised 5 HSV and 4 candida samples, which together with the “Training Set RF” samples made up the complete “Discovery Approach/References” subcohort. 1 additional HSV and candida sample was used for diagnostic testing, which together with the “Test Set (RF)” samples constituted the “Test” subcohort. Every sample was evaluated by experienced pathologists (S.S., K.S., S.M., F.F.). Patient characteristics are summarized in Table EV1A.”

In summary: 20 + 16 + 8 + 5 + 4 + 1 + 1 = 55

We refer to this detailed explanation also in the first paragraph of the Results and Discussion section.

Liquid chromatography-Mass Spectrometry (LC-MS/MS)

Q1

Peptides were eluted and separated on the PharmaFluidics μ PAC nano-LC column: 50 cm μ PAC C18 with a pillar diameter of 5 μ m, inter-pillar distance of 2.5 μ m, pillar length/bed depth of 18 μ m, external porosity of 59%, bed channel width of 315 μ m and bed length of 50 cm; pillars are superficially porous with a porous shell thickness of 300 nm and pore sizes in the order of 100 to 200 Å by a linear gradient from 2% to 30 % of buffer B (80% acetonitrile and 0.08% formic acid in HPLC-grade water) in buffer A (2% acetonitrile and 0.1% formic acid in HPLC-grade water) at a flow rate of 300 nl per min. The remaining peptides were eluted by a short gradient of 10 minutes from 30% to 95% buffer B; followed by 25 minutes at 2% of buffer B, the total gradient run was 120 min.

*FOR HOW LONG was the LC gradient developed? (120-10-25=85 min?)**

We appreciate this comment. While the total run time was 120 min, the gradient length was 85 minutes from 2-30% buffer B, as correctly calculated by the reviewer. We have now included this information in the methods section.

Q2

There is no explanation of the "two BSA washes were used between samples". Please provide a reference or details (e.g. 1% ??) , as not all researchers know this 'trick'.

BSA QC runs are widely and commonly used in LC-MS tandem mass spectrometry analyses in proteomics labs and core facilities. The BSA proteins digest plays several roles: it allows monitoring of metrics such as retention time reproducibility, LC column equilibration following a TRAP wash with high acetonitrile and it reduces non-specific binding and peptides adsorption in the flow path within the LC system, thereby minimizing carry-over.

(e.g. www.sciencedirect.com/science/article/pii/S0003267015010053#abs0010)

The amount of BSA digest injected is 5 femtomole. This is now described in the methods section.

Data analysis

Q3

...the 11 iRT peptide sequences were manually added to **the database and used during DIA-NN search to generate the precursor ion library used for MS data analysis. **

*For a reference to the database please add see *Data analysis and statistics**

...the 11 iRT peptide sequences were manually added to **the database (see below Data analysis and statistics) and used during DIA-NN search to generate the precursor ion library used for MS data analysis. **

iRT standard peptides are provided as protein FASTA sequence and manually added to the Human Uniprot FASTA sequence used for DIA-NN search. Thus, subjected to the same settings used for precursor ion library generation. iRT standards were only spiked into samples to allow control of retention time (RT) across all sample analyses. This is an extra step of QC to ensure that there is no RT shift happening during the MS samples processing that could affect the identification and quantification process. Further, the iRT peptide measured RT were compared and plotted to their linear RT to assess the quality of the MS runs and the performance of DIA-NN search. This information has been added to the methods section.

Q4

LC-MS/MS was performed at the Proteome Center Tuebingen (PCT). Here, 250 ng of peptides were loaded onto an Easy-nLC 1200 system coupled to a quadrupole Orbitrap Exploris 480 mass

spectrometer (all Thermo Fisher Scientific, Waltham, MA, USA) as previously described (Krauss et al, 2023).

*Shouldn't reference of Kraus be omitted? *(Krauss et al, 2023).**

We thank the reviewer for his/her careful reading and totally agree that the reference "Krauss et al." was mistakenly still included in the references, which we have now removed. However, we believe that the entire sentence that the reviewer refers to "LC-MS/MS was performed at the Proteome Center Tuebingen (PCT)..." was actually no longer present in the revised manuscript. It is possible that, due to the extensive markups and tracked changes, it appeared to the reviewer that this sentence was still included.

Q5

It is not clear to me why iRTs used when DIA-NN was used (in library free mode) as a search engine? I anticipate others (e.g. Rochman group) are keen to compare this work with their own, so would encourage all steps to be described precisely.

iRT sequences were used in library free mode, to check the identification performance of LC-MS and DIA-NN search and to control retention time stability. The measured RTs of the iRT peptides were compared and plotted to their linear RTs to assess the quality of the MS runs and the performance of DIA-NN search. This information has been added to the methods section.

Q6

I do not understand the reporting here. Some of the text seems superfluous:

The article states that DIA-NN v1.8.1 is used and then go about explaining the steps the program performs, whereas these steps are merely 'clickable' selections in the DIA-NN GUI. The way it is written here, it seems like these are independent steps by the user. For example: DIA-NN allows the choice of one's own library, or a FASTA formatted library (from PRIDE or SwissProt, e.g.). The authors state:

"First, a precursor ion library was generated using FASTA digest for library-free search in combination with deep learning- based spectra prediction. An experimental library generated from the DIA-NN search was used for cross-run normalization and mass accuracy correction. Only high-accuracy spectra with a minimum precursor FDR of 0.01, and only tryptic peptides (2 missed Tryptic cleavages) were used for protein quantification. "

The DIA-NN search can be performed either, by loading the FASTA files and the MS files simultaneously for search, or by first generating the spectral library separately and then performing the search by using the spectral library. We provided a detailed description of our computational process to clearly explain the strategy we used, as it is recommended to first use DIA-NN to generate the precursor library from the FASTA file and then do the search by loading the spectral library, and MS files (see also DIA-NN github page: <https://github.com/vdemichev/DiaNN>).

The details about using the experimental library generated for cross-run normalization and mass accuracy correction are provided as other users of DIA-NN could set a specific mass accuracy (20 ppm is widely used), rather than allowing DIA-NN to perform the correction, and turn off the option of cross-run normalization.

Taken together, after addressing the comments from Reviewer #1 and Reviewer #2 as well as considering the responses mentioned above and the information added to the re-revised manuscript, along with the details attached to the raw data deposited in the MassIVE repository, we strongly believe that our work is reproducible. We have made every effort to clarify the Methods section and correct any inaccuracies, so that all necessary information and details provided are specific to our experimental procedures and important to reproduce our results by the scientific community.

6th Dec 2024

Dear Dr. Singer,

Thank you for the submission of your revised manuscript to EMBO Molecular Medicine. We have now received the reports from the referees that were asked to re-assess it. As you will see, while Reviewer #4 finds your study suitable for publication, Reviewer #3 continues to have concerns and remains unconvinced of the clinical impact of your study. We also reached out to an expert advisor to evaluate Reviewer 3's previous concerns and your response to them, who felt that the concerns had been sufficiently addressed and in general that the comments did not challenge the main conclusions of the study. Based on their advice and discussion of all of the reports (included below) within our editorial team, I am pleased to inform you that we will be able to accept your manuscript pending the following final amendments and appropriate response to reviewers:

- 1) In the Data availability section please include the URLs that direct to the specific dataset (just after the accession code). The proteomics datasets should now be made publicly available.
- 2) Author contributions: Please remove it from the manuscript and specify author contributions in our submission system. CRediT has replaced the traditional author contributions section because it offers a systematic machine-readable author contributions format that allows for more effective research assessment. You are encouraged to use the free text boxes beneath each contributing author's name to add specific details on the author's contribution. More information is available in our guide to authors:
<https://www.embopress.org/page/journal/17574684/authorguide#authorshipguidelines>
- 3) In the Methods, please take care of the following:
 - Please rename "Material and Methods" to "Methods"
 - Studies with human research participants: The use of human samples requires informed consent, which you mention in the Author Checklist has been given. Please also include a statement in the relevant methods section that written, informed consent from patients was acquired.
 - Please also state in this section that the experiments conformed to the principles set out in the WMA Declaration of Helsinki and the Department of Health and Human Services Belmont Report. Please note that this is a separate statement from the specific ethics committee approval and informed consent.
 - Please ensure that a statement on whether or not blinding was done is included in the Methods even if no blinding was done.
- 4) All materials and methods need to be described in the main text using our 'Structured Methods' format, which is required for all research articles. According to this format, the Methods section includes a Reagents and Tools Table (listing key reagents, experimental models, software and relevant equipment and including their sources and relevant identifiers) followed by a Methods and Protocols section describing the methods using a step-by-step protocol format. The aim is to facilitate adoption of the methodologies across labs. More information on how to adhere to this format as well as a downloadable template (.docx) for the Reagents and Tools Table can be found in our author guidelines:
<https://www.embopress.org/page/journal/17574684/authorguide#structuredmethods>
An example of a Method paper with Structured Methods can be found here:
<https://www.embopress.org/doi/10.15252/msb.20178071>.
- 5) The Results and Discussion should be separate sections.
- 6) Please place individual sections of the manuscript in the following order: Title page - Abstract & Keywords - Introduction - Results - Discussion - Methods - Data Availability - Acknowledgements - Disclosure and Competing Interests Statement - The Paper Explained - References - Figure Legends - Expanded View Figure Legends.
- 7) For the figures and figure legends, please take care of the following:
 - Please note that the box plot needs to be defined in terms of minima, maxima, centre, bounds of box and whiskers, and percentile in the legend of figure 7d.
 - Please note that information related to n is missing in the legend of figure 7d.
 - Please note that for heatmap present in figures EV 3a; EV 5b; a numbered scale bar is not provided. This needs to be rectified.
- 8) Please rename the Tables EV1-EV7 to Dataset EVs. Each excel file can consist of multiple tabs with each tab containing a separate sub-table/panel (so Table EV1A, B, and C can all be incorporated into a single excel file). Each dataset will need to have a legend also added to the corresponding file in a separate tab. Please also be sure to update their callouts in main manuscript text.
- 9) Callouts in the manuscript are missing for the current Dataset EV1 and Code EV1. A legend for the current Dataset EV1 should also be incorporated into the excel file in a separate tab. A readme file should be included for Code EV1 as a legend/instructions to help readers in running the code.
- 10) Please remove the abbreviations list from the manuscript text. Abbreviations should be defined in brackets after their first mention in the text, not in a list of abbreviations. As for the list of gene names, this section should also not be in the manuscript, but you may want to consider including this information in an EV table.
- 11) Funding: Please note that funding information should be given in the "Acknowledgements" section (not in its own separate section).
- 12) Synopsis:
 - Synopsis image: currently the text and figures in the bottom left two panels of the synopsis image are very small and difficult to read at the actual size of the image (which should remain as is at 550 pixels wide x (300-600) pixels high). Please consider

whether you would like to increase the text or image size for readability.

- Synopsis text: Please shorten the 4th bullet point to a max of 30 words

13) As part of the EMBO Publications transparent editorial process initiative (see our policy here:

https://www.embopress.org/transparent-process#Review_Process), EMBO Molecular Medicine will publish online a Peer Review File (PRF) to accompany accepted manuscripts. This file will be published in conjunction with your paper and will include the anonymous referee reports, your point-by-point response and all pertinent correspondence relating to the manuscript. Let us know whether you agree with the publication of the PRF and as here, if you want to remove or not any figures from it prior to publication. Please note that the Authors checklist will be published at the end of the PRF.

14) Please provide a point-by-point letter INCLUDING my comments as well as the reviewer's reports and your detailed responses (as Word file).

I look forward to reading a new revised version of your manuscript as soon as possible.

Yours sincerely,

Poonam Bheda

Poonam Bheda, PhD
Scientific Editor
EMBO Molecular Medicine

***** Reviewer's comments *****

Referee #3 (Remarks for Author):

In this revised manuscript, the authors have addressed some of the concerns expressed by this reviewer. However, this reviewer is left unconvinced about how this proteomics approach may be truly helpful to distinguish esophagitis conditions of differential etiologies, thus having a substantial impact on clinical practice; and/or providing novel mechanistic insights into the pathogenesis of differential esophagitis conditions.

This work may represent a technological advancement, if not done on other tissue types and diseases, but confirmatory at best without a clear focus of disease or cell types, cellular processes, signaling pathways, or molecules and validation of the screening results by functional assays. None of these were focused and data were not validated experimentally.

The combined Results/Discussion section was written in a very descriptive manner. Even though proteomics data was shown to agree between the training sets and the test sets in supplementary information (EV5), it is unclear whether the authors analyzed any test set samples in a blinded and unbiased manner. If histologic and clinical diagnosis was already known for test set samples, it would not be surprising that the authors got a high concordance between the training sets and the test sets by proteomics. Additionally, the sample size for each esophagitis condition is small (5 or less). In some cases, test sets comprise a single sample only per disease condition (e.g., HSV, borderline EoE), raising a concern over the statistical rigor.

Referee #4 (Comments on Novelty/Model System for Author):

Technical quality: High

This is easy, and it is excellent. The authors report a high diagnostic accuracy, superior to conventional histology alone. Their findings also confirm and correlate well with previous histological results, demonstrating a validation of their methods.

Novelty: High.

This may be the first study to apply this approach to this specific disease. Novelty is apparent in the use of AI combined with high-density proteome analysis, which could be groundbreaking if limited to this disease. However, similar methods are becoming more common, particularly with recent advances in mass spectrometry instrumentation, so it's not entirely unique in a broader context.

Medical impact: High

The potential medical impact is significant, particularly if the findings can be translated into clinical practice. The combination of proteomic and histological approaches could lead to improved diagnostic tools for esophagitis, which would be a big

improvement. However, the true impact will depend on whether clinicians can easily understand and apply these and other future insights. This is where the journal could play a pivotal role - by ensuring that this research is communicated effectively not only within the scientific community but also to patient organisations, clinicians, and potentially regulatory bodies. Such work could ultimately pave the way for personalised approaches to disease diagnosis and treatment.

Adequacy of model system: ?

The limited sample size was discussed here. While the authors acknowledge the need for larger patient cohorts to validate their findings, the small sample size may affect the conclusions of the results eventually. Future studies with higher patient numbers could therefore also include those with complex medical histories. Nevertheless, this should not be seen as a reason to withhold publication.

Overall:

This study presents a promising tool for esophagitis diagnosis, demonstrating improved accuracy over conventional histology. While limitations related to sample size and clinical complexity remain, the importance of publishing this work lies in its potential to advance the field. Continued research will help refine the approach and assess its broader applicability.

Referee #4 (Remarks for Author):

Publish, but not as a short report, please. While I initially found the wording somewhat intricate, I believe we now have a high-quality report that reflects trends we are seeing in other areas of clinical proteomics as well. These findings, approaches, and conclusions should be part of public discourse, as they importantly point toward personalised medicine.

The article demonstrates that by combining proteomic and histological approaches, a promising tool for esophagitis diagnosis exists, offering improved accuracy compared to conventional histology alone.

While limitations remain - primarily concerning sample size and, by extension, the complexity of clinical settings - this signals the need for follow-up on a larger scale.

Overall, the article presents an interesting and valid approach, justifying the direction of research and supporting the notion that more work should be done to refine these findings.

Ideally, this study will inspire others to build on this work before its broader adoption in clinical practice. Collaborating with other researchers to address sample size issues, resource requirements, and further validation would accelerate progress and enhance clinical impact.

***** Expert advisor's comments *****

I read carefully the comments by the reviewer #3 and the authors responses to them, and I truly think that the authors have very well responded to most of the comments and for me none of the questions by the reviewer were such that they would challenge the main conclusions of the study. I however do have three comments that and I think addressing them would further increase the quality and interest of the paper.

Numbers refer to original comments by the reviewer #3:

3. I do think that the PCA plot offers additional insights to the data and increases the confidence that different disease states can be recognized by the approach. Therefore I would include the PCA blot as EV data in the final manuscript

4. As I read the figure legends of 2A and 3A, the DAPs in these Volcano blots represent all differentially expressed proteins, not the eosinophils-related proteins the reviewer #3 was concerned about. If the authors cite as eosinophil-related proteins those that are indicated with protein names in these volcanoblots, then the numbers would be 6 vs. 12, not 6 vs. 9 as indicated in their response.

In order to clarify this comment appropriately, please indicate the criteria for proteins considered as "eosinophil-related" and their respective numbers between 2A and 3A in the results text when describing the results of 3A.

6. This is a valid comment by the reviewer #3, but knowing the intricate challenges with antibody specificities in IHC, validation of relevant number of proteins in each disease state studied would be a significant undertaking. Therefore I would comment this on the "limitations of the study" in the discussions section.

***** Reviewer's comments *****

Referee #3 (Remarks for Author):

In this revised manuscript, the authors have addressed some of the concerns expressed by this reviewer. However, this reviewer is left unconvinced about how this proteomics approach may be truly helpful to distinguish esophagitis conditions of differential etiologies, thus having a substantial impact on clinical practice; and/or providing novel mechanistic insights into the pathogenesis of differential esophagitis conditions.

This work may represent a technological advancement, if not done on other tissue types and diseases, but confirmatory at best without a clear focus of disease or cell types, cellular processes, signaling pathways, or molecules and validation of the screening results by functional assays. None of these were focused and data were not validated experimentally.

The combined Results/Discussion section was written in a very descriptive manner. Even though proteomics data was shown to agree between the training sets and the test sets in supplementary information (EV5), it is unclear whether the authors analyzed any test set samples in a blinded and unbiased manner. If histologic and clinical diagnosis was already known for test set samples, it would not be surprising that the authors got a high concordance between the training sets and the test sets by proteomics. Additionally, the sample size for each esophagitis condition is small (5 or less). In some cases, test sets comprise a single sample only per disease condition (e.g., HSV, borderline EoE), raising a concern over the statistical rigor.

We regret leaving this referee still unconvinced regarding the usefulness of the proposed approach in clinical practice. This does not reflect our experience in the close collaboration with our clinical colleagues who, in fact, highly appreciate AI-assisted morpho-proteomics as a useful adjunct in esophagitis diagnostics and therapeutic decision making. We consider the comparative study design, which the reviewer criticizes as a lack of focus, a prerequisite to identify discriminative and shared proteomic changes to improve diagnostic accuracy. However, we acknowledge that this rather broad approach comes at the expense of mechanistic detail, which is better addressed in studies focusing exclusively on a single disease. We believe that both approaches are valid and should ideally complement each other. This is exemplified by alterations of the MCM complex, which have previously been described and investigated in mechanistic detail by Rochman *et al.* in EoE and linked to basal cell hyperplasia with associated therapeutic implications. We, in turn, observed MCM alterations not only in EoE, but also in GERD and CD because of our comparative approach and thereby suggest the MCM-directed therapy as an option for these esophagitis types as well.

In our understanding traditional blinding should prevent bias and minimize subjective influence or expectations in research. However, machine learning as mathematical system is not considered susceptible to this type of bias as it does not “expect” anything. The concept of supervised machine learning (RandomForest in this study) inherently requires labelled data and is not compatible with

conventional blinding methods. We also would like to point out that in the combined histo-proteomic RF, "histo" does not represent the histological diagnosis, but descriptive histological features. There are in principle two methods to train and validate/test machine learning models: either by using two separate datasets (training and test/validation data set) or by using one dataset that is randomly partitioned with the model trained and tested on these separated data subsets. Based on the request by reviewer#1 in the first round of revision we used the former strategy with two different datasets, which, in fact, increased the total number of (non-infectious) analysed samples per disease or control finally to n=9 (not n=5). Having performed these standard procedures of training and validation, we applied the model to challenging samples in Fig 7C, where the diagnosis was not clear/borderline by conventional histological methods. Here, the algorithm indeed provided support in diagnostic decision making. Finally, we acknowledge the sample size as a limitation in the manuscript and expect future studies with larger cohorts to support/refine our findings.

Referee #4 (Comments on Novelty/Model System for Author):

Technical quality: High

This is easy, and it is excellent. The authors report a high diagnostic accuracy, superior to conventional histology alone. Their findings also confirm and correlate well with previous histological results, demonstrating a validation of their methods.

Novelty: High.

This may be the first study to apply this approach to this specific disease. Novelty is apparent in the use of AI combined with high-density proteome analysis, which could be groundbreaking if limited to this disease. However, similar methods are becoming more common, particularly with recent advances in mass spectrometry instrumentation, so it's not entirely unique in a broader context.

Medical impact: High

The potential medical impact is significant, particularly if the findings can be translated into clinical practice. The combination of proteomic and histological approaches could lead to improved diagnostic tools for esophagitis, which would be a big improvement. However, the true impact will depend on whether clinicians can easily understand and apply these and other future insights. This is where the journal could play a pivotal role - by ensuring that this research is communicated effectively not only within the scientific community but also to patient organisations, clinicians, and potentially regulatory bodies. Such work could ultimately pave the way for personalised approaches to disease diagnosis and treatment.

Adequacy of model system: ?

The limited sample size was discussed here. While the authors acknowledge the need for larger patient cohorts to validate their findings, the small sample size may affect the conclusions of the results eventually. Future studies with higher patient numbers could therefore also include those with complex medical histories. Nevertheless, this should not be seen as a reason to withhold publication.

Overall:

This study presents a promising tool for esophagitis diagnosis, demonstrating improved accuracy over conventional histology. While limitations related to sample size and clinical complexity remain, the importance of publishing this work lies in its potential to advance the field. Continued research will help refine the approach and assess its broader applicability.

Referee #4 (Remarks for Author):

Publish, but not as a short report, please. While I initially found the wording somewhat intricate, I believe we now have a high-quality report that reflects trends we are seeing in other areas of clinical proteomics as well. These findings, approaches, and conclusions should be part of public discourse, as

they importantly point toward personalised medicine.

The article demonstrates that by combining proteomic and histological approaches, a promising tool for esophagitis diagnosis exists, offering improved accuracy compared to conventional histology alone.

While limitations remain - primarily concerning sample size and, by extension, the complexity of clinical settings - this signals the need for follow-up on a larger scale.

Overall, the article presents an interesting and valid approach, justifying the direction of research and supporting the notion that more work should be done to refine these findings.

Ideally, this study will inspire others to build on this work before its broader adoption in clinical practice. Collaborating with other researchers to address sample size issues, resource requirements, and further validation would accelerate progress and enhance clinical impact.

We are most grateful for the supportive comments provided by the reviewer and greatly appreciate the very positive feedback on the revised version of our manuscript!

***** Expert advisor's comments *****

I read carefully the comments by the reviewer #3 and the authors responses to them, and I truly think that the authors have very well responded to most of the comments and for me none of the questions by the reviewer were such that they would challenge the main conclusions of the study. I however do have three comments that and I think addressing them would further increase the quality and interest of the paper.

We sincerely thank the expert advisor for taking the time to review our replies to the comments and concerns of Referee #3 and for acknowledging our efforts to respond to them adequately.

Numbers refer to original comments by the reviewer #3:

3. I do think that the PCA plot offers additional insights to the data and increases the confidence that different disease states can be recognized by the approach. Therefore I would include the PCA blot as EV data in the final manuscript.

As suggested by the expert advisor we have included the PCA plot as Fig. EV5 and refer to it in the Results section where we introduce the machine learning approach. The PCA plot shows good separation particularly between normal and EoE samples. However, it also reveals a high similarity and thereby poor discrimination between the GERD and CD cases being clustered together. This finding supports the use of supervised machine learning to better distinguish between all esophagitis types and also enables the incorporation of histological features.

4. As I read the figure legends of 2A and 3A, the DAPs in these Volcano blots represent all differentially expressed proteins, not the eosinophils-related proteins the reviewer #3 was concerned about. If the authors cite as eosinophil-related proteins those that are indicated with protein names in these volcanoblots, then the numbers would be 6 vs. 12, not 6 vs. 9 as indicated in their response. In order to clarify this comment appropriately, please indicate the criteria for proteins considered as "eosinophil-related" and their respective numbers between 2A and 3A in the results text when describing the results of 3A.

There may be a misunderstanding between the expert advisor's and our interpretation of the

respective Reviewer#3's comment: *'The authors state that data indicated both EoE and GERD samples had eosinophilia. It is unclear to what extent eosinophils-related proteins are differentially expressed between EoE and GERD. It would be surprising if most GERD showed eosinophils-related proteins to an extent that could not be distinguished from EoE.'*

We interpreted this to mean that the reviewer would find it surprising if the expression levels of eosinophilic markers (e.g. EPX, RNASE2, RNASE3) were not higher in the EoE samples compared to the GERD samples. In our response, we therefore referred to the fold change (FC) of eosinophilic markers detected in both conditions. Accordingly, we pointed out that in the respective volcano plots the log₂ FC of the eosinophilic markers EPX, RNASE2, and RNASE3 is ~8.5 in EoE (Fig.2A) and ~6 in GERD (Fig. 3A). This means that there is higher abundance of the eosinophilic markers EPX, RNASE2, and RNASE3 in EoE compared to GERD, as expected. The labelling of the proteins in the respective volcanos was not intended to only show eosinophils-related proteins, but refer to the specific context in each figure. In Figure 2A it highlights the top 10 most upregulated proteins in EoE which were used for the metascape analyses in Figure 2C and which included (but were not entirely composed of) eosinophilic markers as illustrated in Figure 2B. In (former) Figure 3A (and 3B) we refer to the composition of the immune infiltrate in GERD that contains neutrophils with the markers CTSG and ELANE and eosinophils with the markers EPX, RNASE2, RNASE3 and PRG2. In order to avoid any confusion, we highlighted all significantly altered eosinophilic markers (EPX, RNASE2, RNASE3, PRG2, **CLC**, and **PRG3**) in the updated Figures 3A and 3B. Moreover, we have included an additional sentence in the Results section referring to Figure 3A and 3B describing which markers are eosinophils-related. Finally, to be as consistent as possible throughout the figures we also included CLC as additional significantly altered eosinophilic marker in Fig 4A and 4B. Interestingly PRG3 was not significantly changed in CD and therefore is not highlighted.

6. This is a valid comment by the reviewer #3, but knowing the intricate challenges with antibody specificities in IHC, validation of relevant number of proteins in each disease state studied would be a significant undertaking. Therefore I would comment this on the "limitations of the study" in the discussions section.

We appreciate expert advisor's understanding of the efforts and challenges to establish and validate IHC markers for a relevant number of proteins. We have included a paragraph in the Discussion section pointing out the limitation of having exclusively used proteomics in our study. This paragraph also highlights that a subset of markers has already been detected by other groups using alternative methods (eg. IHC, IF, or IB). Since these markers refer to EoE that paragraph is integrated in the discussion of the EoE data.

14th Jan 2025

Dear Dr. Singer,

Congratulations on an excellent manuscript, I am pleased to inform you that your manuscript has been accepted for publication in the EMBO Molecular Medicine.

Yours sincerely,

Poonam Bheda, PhD
Scientific Editor
EMBO Molecular Medicine
